# Polymorphic amyloid nanostructures of hormone peptides involved in glucose homeostasis display reversible amyloid formation

Dániel Horváth [1], Zsolt Dürvanger [1,2], Dóra K. Menyhárd [1,2], Máté Sulyok-Eiler[2,3], Fruzsina Bencs [2,3], Gergő Gyulai [4], Péter Horváth [5], Nóra Taricska[1] & András Perczel [1,2] ✉

A large group of hormones are stored as amyloid fibrils in acidic secretion vesicles before they are released into the bloodstream and readopt their functional state. Here, we identify an evolutionarily conserved hexapeptide sequence as the major aggregation-prone region (APR) of gastrointestinal peptides of the glucagon family: xFxxWL. We determine nine polymorphic crystal structures of the APR segments of glucagon-like peptides 1 and 2, and exendin and its derivatives. We follow amyloid formation by CD, FTIR, ThT assays, and AFM. We propose that the pH-dependent changes of the protonation states of glutamate/aspartate residues of APRs initiate switching between the amyloid and the folded, monomeric forms of the hormones. We find that pH sensitivity diminishes in the absence of acidic gatekeepers and amyloid formation progresses over a broad pH range. Our results highlight the dual role of short aggregation core motifs in reversible amyloid formation and receptor binding.

An increasing number of physiological roles have been revealed and associated with functional amyloids[1]. The primary task of these functional fibrils was thought to be limited to the maintenance of the structural integrity of the cell, adhesion, or invasion via biofilm formation[2–4]. Recent studies shed light on their importance in signalization, as well as in fine-tuning protein functions[5–7]. Furthermore, dozens of peptide hormones of the endocrine system are amassed in acidic secretory vesicles as structured aggregates or amyloids[8,9]. This compartmentalized reservoir system allows for storing polypeptides selectively in a dense, phase-separated, and stabilized form. Triggered by the secretion signal, the matured granules immediately release the stored hormones into the bloodstream, creating an acute export beyond the biosynthetic capacity[10]. Fibrils dismantle due to the changing chemical environment[11], and the nascent monomers readopt their physiologically functional forms. As of yet, among these endocrine hormones, the structure of only three amyloid-states, those of glucagon[12,13] and β-endorphin[14] have been determined at atomic resolution.

Glucagon (gluc), glucagon-like peptide 1 and 2 (GLP-1 and GLP-2), glucose-dependent insulinotropic peptide (GIP), and their orthologue exendins[15] are members of the larger family of secretin-like gastrointestinal hormones[16] (Fig. 1a). Following their biosynthesis, the (pre) prohormones are processed by tissue-specific prohormone convertases during maturation in the Golgi apparatus. Hormones of the

[1]ELKH-ELTE Protein Modeling Research Group ELTE Eötvös Loránd University, Pázmány Péter sétány 1/A, Budapest H-1117, Hungary. [2]Laboratory of Structural Chemistry and Biology ELTE Eötvös Loránd University, Pázmány Péter sétány 1/A, Budapest H-1117, Hungary. [3]Hevesy György PhD School of Chemistry, ELTE Eötvös Loránd University, Pázmány Péter sétány 1/A, Budapest H-1117, Hungary. [4]Laboratory of Interfaces and Nanostructures, Institute of Chemistry, Eötvös Loránd University, Pázmány Péter sétány 1/A, Budapest H-1117, Hungary. [5]Department of Pharmaceutical Chemistry, Semmelweis University, Hőgyes Endre utca 9, Budapest 1092, Hungary. ✉e-mail: perczel.andras@ttk.elte.hu

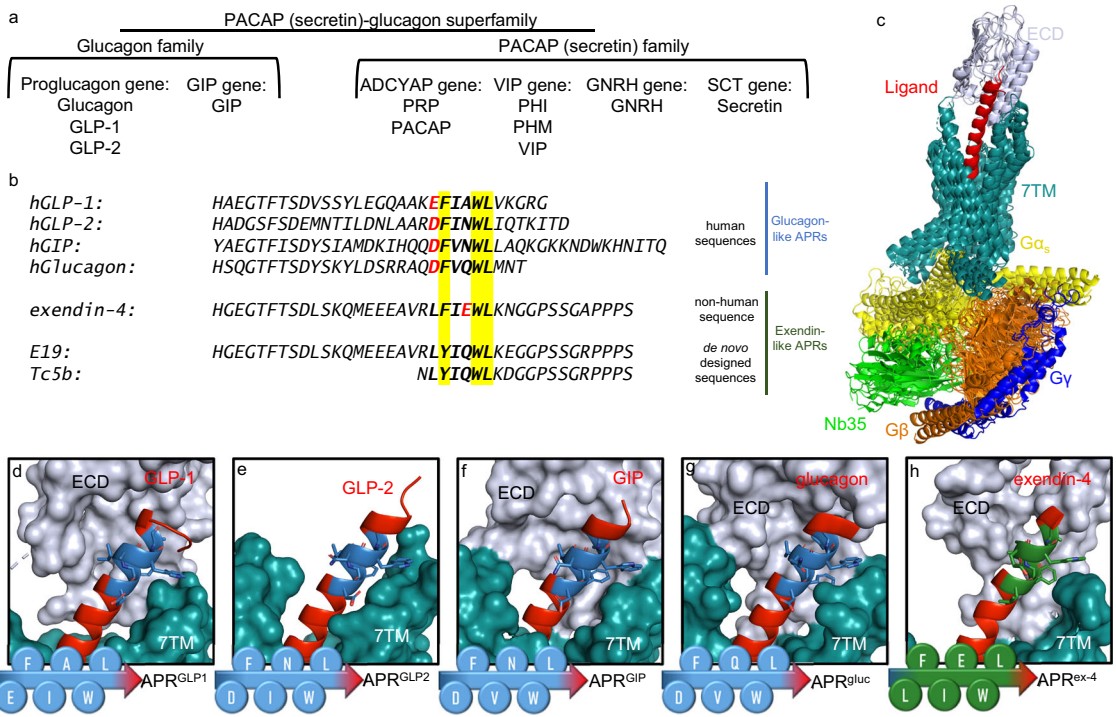

**Fig. 1 | Overview of human secretin-like hormones. a** Mammalian genomes contain 6 genes that encode 10 secretin-like hormones. GLP-1 glucagon-like peptide 1, GLP-2 glucagon-like peptide 2, GIP gastric inhibitory polypeptide, ADCYAP adenylate cyclase activating peptide, PACAP pituitary adenylate cyclase activating protein, PRP PACAP-related peptide, PHI peptide histidine isoleucine, PHM peptide histidine methionine, VIP vasoactive intestinal peptide, GNRH growth hormone releasing hormone, SCT secretin. Orthologues of the gene encoding exendins were found in various vertebrates, but not in mammals. **b** Primary sequences of the (human) secretin-like hormone polypeptides share high similarity despite their distinct physiological roles. (Glucagon-family hormone sequences of the different vertebrate species are compared in Supplementary Fig. 1) The evolutionarily conserved residues in the aggregation-prone regions (APR) of the investigated peptides are highlighted in yellow. Acidic residues ($pK_a < 5$) are depicted in red. All peptide ligands adopt a partially disordered/nascent helix in the solution along the whole sequence (PDB entries are as follows GLP-1: 1D0R; GLP-2: 2L63; GIP: 2OBU; glucagon: 1GCN; exendin-4: 1JRJ; E19: 2MJ9; Tc5b: 1L2Y). **c** Superimposed ligand-receptor complexes reveal a common receptor-binding mechanism in the glucagon-like subfamily of the Class B1 GPCRs. The segment close to the C-terminal of the ligands recognizes the binding surface of the receptor's extracellular domain, while the also highly homologous N-terminal segment of the already bounded ligand occupies the cavity formed by 7TM helices inducing conformational changes in the helices, which activate the receptor. **d–h** The highlighted ligand-APR sequences interact at identical positions with the extracellular domain of their respective receptor in the hormone ligand-receptor complexes. (PDBs: (**d**): GLP-1-GLP1R: 5VAI; (**e**): GLP-2-GLP2R: 7D68; (**f**): GIP-GIPR: 7DTY; (**g**): Glucagon-GCGR: 6LMK; (**h**): exendin-4-GLP1R: 7LLL). The conserved aromatic residues play a crucial role in partner recognition via aromatic interactions. The APR hexapeptide segments are also illustrated as schematic β-strands (without any terminal protection), and termed as APR$^{GLP1}$, APR$^{GLP2}$, APR$^{GIP}$, APR$^{gluc}$, APR$^{ex-4}$, and APR$^{Tc5b}$. Glucagon-like peptides (blue) carry negatively charged side chains at the first position of their APR segments, while exendin-4 sequence (green) has it in the middle of its APR. The synthetic exendin-like APR, LYIQWL, does not contain acidic side chains.

glucagon family have a common evolutionary origin that is reflected in high homology among their primary sequences in all vertebrate species[17] (Fig. 1b and Supplementary Fig. 1). Despite their close evolutionary relationship, the major actions of these hormones are non-overlapping, covering the regulation of glucose homeostasis, appetite, and maintaining the proper function of intestines[18,19]. Their synthetic derivatives are successfully applied in the therapy of glucose metabolism-related disorders, with growing interest in further indications[20]. Each hormone peptide has its own class B type G-protein coupled receptor[19,21] functioning along identical receptor activating schemes (Fig. 1c).

The amphiphilic receptor-binding segments of these essentially helical ligands, which provide for selective and specific receptor-ligand interactions, all contain a highly conserved xFxxWL hexapeptide sequence that may also be the basis of multiagonism[20] (Fig. 1d–h). In this work, we show that these hexapeptides are not only involved in the binding to the extracellular domain of their respective receptors but play a prominent role as primary aggregation-prone regions (APRs) too[8] (Supplementary Fig. 1) in vesicles-located amyloid formation. We carried out a detailed examination of the amyloidic nature and its pH-dependent reversibility in the case of the APRs derived from the human GLP-1, GLP-2, glucagon/oxyntomodulin and GIP, as well as the

non-mammalian exendin-3 & 4 and their de novo designed derivatives of Trp-caged fold, those of Tc5b and E19[22–24] (Fig. 1b). In total, we present 9 amyloid-like, high resolution crystal structures of 4 hexapeptides, including polymorphic[25] variants.

## Results

### Amyloid formation typically requires acidic conditions

The aggregation propensities of hexapeptides derived from the gastrointestinal hormones were monitored by Circular Dichroism (CD) spectroscopy in the far-UV range. The pH values were set to cover the entire range of charge distribution accessible for each peptide. (Supplementary Figs. 3–9) Due to the reduced solubility and inherent aggregation potential of these hydrophobic hexapeptides near their isoelectric points, measurement near the corresponding pH range was challenging, even though the concentration used for spectroscopy ($c ≈ 0.2$ mM) was typically two orders of magnitude lower than that of the hormones within the granules ($c ≈ 30$ mM)[10]. Hormone derived natural APRs dissolved completely at physiological pH and above, presenting a U-type CD spectra characteristic of the unfolded state, which remained unchanged even after weeks of incubation time. On the contrary, under acidic conditions, β-type spectra were detected in the case of APR$^{ex-4}$, APR$^{Tc5b}$ and/or atypical β′-type for APR$^{GLP1}$, APR$^{GLP2}$,

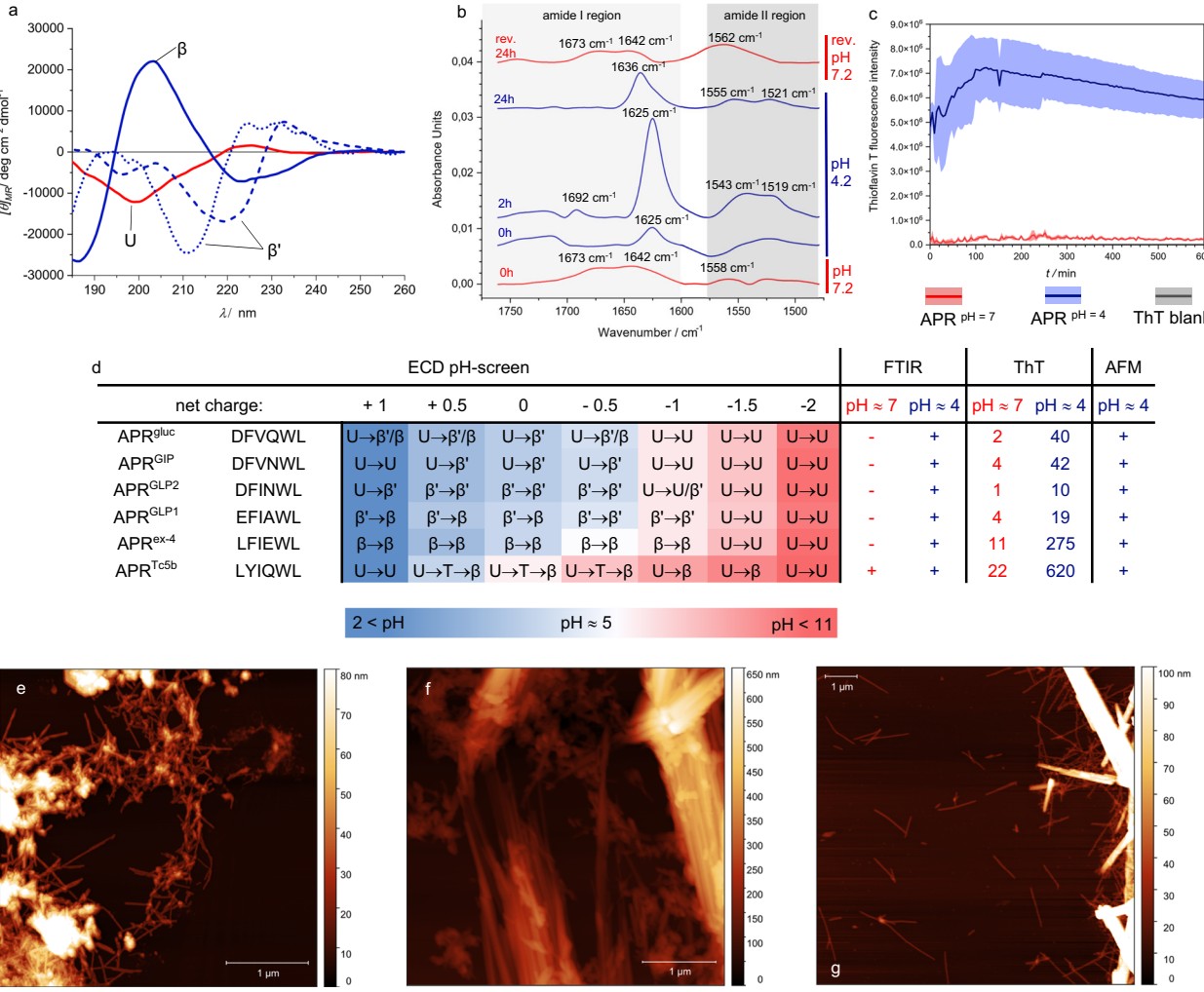

**Fig. 2 | Orthologous techniques used to characterize the amyloid-like fibril formation of the APRs. a** Characteristic circular dichroism (CD) spectra detected during the amyloid formation of glucagon-like APRs. (Supplementary Figs. 3–9 shows all CD spectra in detail.) U: U-type CD-spectrum with a negative maximum at -200 nm characteristic to the unfolded state. β: Exendin-like sequences gave a typical B-type CD-spectrum, characteristic for β-strands (a large positive maximum at -205 nm, with a smaller (optional) negative maximum at -225 nm). β': Unconventional B-type CD spectrum characterizing the glucagon-like sequences, where the intensive negative maximum at either 205–210 or 220–225 nm, is followed by a positive maximum above 225–230 nm. **b** FTIR spectra of APR$^{GLP2}$ shortly after setting the pH neutral (red) to acidic (blue) reveals an enhanced signal in the amide 1 band region, characteristic of the stacked β-sheets of amyloid-like fibrils. This vibration immediately diminishes when setting the pH back to neutral (top FTIR

spectrum - rev. refers to reversibility). **c** The ThT fluorescence measurement of APR$^{ex4}$ shows an almost thirtyfold increase in intensity in the case of the acidic sample (blue) compared to the neutral sample (red), which can be considered ThT-negative. The darker lines represent the averages, while the brighter belts indicate the standard deviations of three parallel measurements. **d** Overview of the detected signals in relation to the presence of amyloid fibrils. CD spectral shifts indicate secondary structure changes of the hexapeptides as a function of the net charge. The values in the ThT columns show the measured intensity increase ratio compared to the blank ThT sample, which is considered as 1. AFM images provide morphological insight into the assembled fibrillar nanostructures of (**e**) APR$^{gluc}$ ($pH \approx 4$), (**f**) APR$^{GIP}$ ($pH \approx 4$), and (**g**) APR$^{Tc5b}$ ($pH \approx 7$) after 24 h of agitation. Note, the same matured amyloid samples were used for FTIR and AFM measurements. Supplementary Figs. 10–15 collect all the results for each APR individually.

APR$^{GIP}$, APR$^{gluc}$ immediately or shortly after sample preparation: the transition from the initial U-type curves to β/β'-types took, at most, a few hours. (Fig. 2, Supplementary Figs. 10–15a panels) The appearance of the β/β'-type CD spectra under acidic conditions confirmed that all 6 APRs had undergone secondary structure changes, even though the emerging oversized oligomers became gradually undetectable due to the wavelength limitations of CD spectroscopy.

To verify whether the observed non-conventional β-type spectra can be confidently attributed to β-sheet formation and to avoid the light scattering limitations mentioned above, we conducted Fourier Transform Infrared Spectroscopy (FTIR) measurements in aqueous solution at both pH 4.2 and 7.2. By adjusting the pH from 7.2 to 4.2, we detected a strong signal in the amide 1 region, specifically in the 1625–1630 cm$^{-1}$ wavelength range[26,27], indicating the formation of extended β-sheets. In addition, a less intense signal at 1692 cm$^{-1}$ was

also detected, suggesting the presence of antiparallel β-sheet assembly (Fig. 2 Supplementary Figs. 10–15c panels).

To provide additional evidence of the presence of amyloid fibrils, we applied Thioflavin T (ThT) binding assays and used Atomic Force Microscopy (AFM) to determine the morphology of the fibrils. At the acidic pH range, the ThT tests clearly showed the formation of amyloid fibrils for APRs of exendin-4, glucagon, GIP, and Tc5b within the same time interval as observed in CD and FTIR. APR$^{ex-4}$ and APR$^{gluc}$ maintained their increased fluorescence over time, while the latter two APRs exhibited rapid decay. On the other hand, APRs of GLP-1 and GLP-2 showed only a slight increase in fluorescence intensity - but still exceeded, by a factor of more than ~4 and~8, those detected at neutral pH, respectively, suggesting amyloid formation in these cases too (Fig. 2c, d, Supplementary Figs. 10–15ThT panels). The AFM analysis of FTIR samples under acidic conditions after 24 h of agitation showed

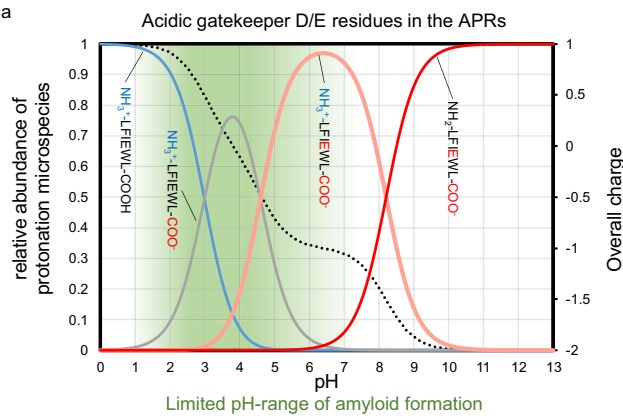

**Fig. 3 | Comparison of the aggregation propensity in light of the charge distribution of the APRs.** The total charge (z - black dotted line - right vertical axis) and the calculated relative abundance of differently charged APR microspecies as a function of the pH are displayed (**a**) in the presence, and (**b**) in the absence of an Asp/Glu acidic gatekeepers. The green background indicates the pH range where the amyloid formation was detected by CD spectroscopy.

the presence of fibrillar assemblies of different sizes and complexity levels, even in the case of APR$^{GLP1}$ and APR$^{GLP2}$, where ThT results were ambiguous. The fibrils or nanotubes composed of β-sheets[28] had a linear, rod-like shape, and the height of individual (proto)filaments ranged from 3.5 to 20 nm. These single filaments were seen to associate in a variety of ways, including tangling up to display periodicity along their longitudinal axis or simply bundling linearly together. Amorphous aggregates and potential nanocrystals could be observed, although nanocrystals smaller than 500 nm in diameter are difficult to distinguish from filaments at this resolution (Fig. 2e–g, Supplementary Figs. 10–15AFM panels).

The U↔β/β′-type CD spectral shift of the hexapeptides takes place in the pH range that corresponds to the change in the protonation state of Asp and Glu side chains. Our results confirm that if the APR contains such a negatively charged gatekeeper[29] residue, then its ability to form amyloid is strictly coupled to the protonation state of the gatekeeper side chains (Fig. 3a, Supplementary Figs. 3–9a panels). When the protonated form becomes predominant, β-sheeted self-association occurs. However, above $pH = 5$, where the deprotonated form is dominant and each of these acidic side chains holds a net negative charge, the solubility of the hexapeptides is enhanced to such an extent that the equilibrium gets shifted toward the disassembled and unstructured monomeric forms. This demonstrates the effective gatekeeper function of both Asp and Glu residues, under a wide range of physiological milieu.

The pH-dependent behavior of the de novo designed APR$^{Tc5b}$ with a primary structure of LYIQWL provided further proof for the gatekeeper concept. Replacing the pH-sensitive molecular switch function of acidic gatekeeper residues by their amide counterpart (E→Q) within the APR$^{Tc5b}$ allows amyloid formation over a much broader pH range, even above $pH = 7$. The pH scan followed by CD indicated that predominantly unipolar states, which usually occur at pH values below 2 and above 11, did not lead to amyloid formation. However, between these two extreme pH conditions, amyloid formation was observed with all techniques used (Fig. 2d, Supplementary Fig. 15).

**Amyloid formation of Glu/Asp comprising APRs can be reversed under physiological conditions**

We found that the formation of matured amyloids that contain the Asp/Glu pH-dependent molecular switch could be reversed. As soon as pH is shifted from acidic to a neutral (physiological) value, amyloid aggregates of gatekeeper-containing APRs readopt their monomeric and unfolded states. (Supplementary Fig. 17) The solutions of the APRs, including those with initial β/β′-type CD spectra, became immediately clear as the precipitate dissolved. The recorded U-type CD spectrum

and the absence of the FTIR amid I band characteristic of β-sheets indicated the presence of a highly dynamic solvated backbone structure. These features remained unchanged for days, supporting that a prompt and complete restoration of the monomeric state took place (Supplementary Figs. 10–15b, c).

To find out whether full-length hormones also form amyloids in a pH-dependent manner, we studied the amyloid formation of glucagon. While there are several examples in the literature that demonstrate the reversible amyloid formation of secretory hormones[8,14], there is no evidence that pertains to the glucagon superfamily. Therefore, we chose to study pre-treated glucagon fibrils and monitored their reversibility in vitro. The results from the combination of CD and AFM experiments confirmed the reversibility of amyloid formation, but a 24-h period was necessary for almost complete recovery (Supplementary Fig. 18). However, additional conditions may enhance the monomerization process in vivo, such as the significant dilution factor at release or the influence of shear forces on the fibrils in the bloodstream.

As shown above, APR$^{Tc5b}$ does not contain gatekeeper residues and is thus capable of amyloid formation over a broad range of pH ($2 < pH < 11$). Recovering the monomeric state from the fibrils of LYIQWL requires extreme pHs (Supplementary Fig. 15b–d). This reversibility can be attributed to the presence of exclusively monopole states—either completely protonated ($pH < 2$) or deprotonated ($pH > 11$) (Fig. 3b). In contrast to the wild-type sequences discussed above, the self-assembly/disassembly of the APR$^{Tc5b}$ amyloid is governed mainly by the improper charge balance of the backbone terminals, and not by switchable acidic side chains. Such reversibility occurring at extreme pH ranges has less biological relevance, although it adheres to the same principle.

Pushing the pH to such extreme values causes an increase in the ionic strength of the solution, which in itself could inhibit or reverse amyloid formation via the effective shielding of the charged intermolecular interactions[30]. Thus, in a control experiment, we attempted to inhibit the amyloid formation of APR$^{Tc5b}$ at $pH = 5.7$, by adding different amounts of salt (5 or 15 mM NaCl) to the samples. The increased ionic strength perturbed amyloid formation and resulted in a decrease of β-type CD-spectral intensity, however, the U-type CD spectra were not regained, indicating that amyloid formation cannot be reversed solely by altering the salt concentration (Supplementary Fig. 19).

**APRs of homologous sequences form amyloid-like crystals of different architecture**

Crystallization of all the studied polypeptides was attempted using the same set of conditions. Amyloid-like single crystals were successfully

**Table 1 | Overview of the amyloid-like APR crystal structures**

| APR | Polymorph | PDB ID | Crystallization conditions | Backbone conformer(s)[a] | Co-crystallising reagent(s) | Amyloid class[b] | Shifted assembly mode[c] |
|---|---|---|---|---|---|---|---|
| DFINWL (GLP-2) | A | 8ANJ | 70% $H_2O$ / 30% ACN, 0.1% TFA, 37 °C at acidic conditions (*pH*: 2–3) | A1 (skyblue) | $H_2O$ | class 1 parallel | Slight lateral shift between every second sheet |
| pEFIAWL (GLP-1) | A | 8ANK | 70% $H_2O$ / 30% ACN, 0.1% TFA, 37 °C at acidic conditions (*pH*: 2–3) | A1 (deepsalmon) | $H_2O$ | class 7 antiparallel | No shift |
| Ac-EFIAWL (GLP-1) | A | 8ONQ | 0.1 M citrate buffer, (*pH*: 4.0) | A1 (aquamarine) A2 (deepblue) | $H_2O$ | class 7 antiparallel | Lateral shift between sheets |
| LFIEWL (exendin-4) | A | 8ANN | 70% $H_2O$ / 30% ACN, 0.1% TFA, 37 °C at acidic conditions (*pH*: 2–3) | A1 (greencyan) | $H_2O$ | class 8 antiparallel | No shift |
| | B | 8ANL | $H_2O$, 37 °C, *pH*: 5.5 | B1 (green) B2 (limon) B3 (darkyellow) B4 (forestgreen) | $H_2O$ | class 7 / 8 antiparallel | Lateral shift between sheets |
| LYIQWL (Tc5b, E19) | A | 8ANH | 66% $H_2O$ / 33% ACN, 0,1% TFA, 4 °C, at acidic conditions (*pH*: 2–3) | A1 (lightpurple) A2 (deeppurple) | $H_2O$, ACN, TFA | class 8 antiparallel | Shifted along the β-spine axis |
| | B | 8ANM | $H_2O$, 37 °C, at isoelectric point of the peptide (*pH*: 5.6) | B1 (teal) | $H_2O$ | class 8 antiparallel | Lateral shift between sheets |
| | C | 8ANI | 90% $H_2O$ / 10% EtOH, 37 °C at acidic conditions (*pH*: 3–5) | C1 (orange) C2 (olive) | $H_2O$, EtOH, TFA | class 8 antiparallel | Shifted along the β-spine axis |
| | D | 8ANG | 70% $H_2O$ / 30% EtOH, 37 °C, at acidic or neutral conditions (*pH*: 4–7) | D1 (red) | EtOH | class 1 / 4 parallel | Lateral shift between sheets |

[a]According to Fig. 4.
[b]Amyloid classes are defined based on Eisenberg classification[32].
[c]Adjacent β-sheets may shift perpendicular to the fibril axis, showing laterally displaced contacting surfaces. β-strand dimers can displace along the β-spine axis, disrupting its axial continuity.

grown from solutions of the APRs of GLP-1, GLP-2, exendin-4, and Tc5b. When crystallized from different solvent mixtures (Table 1), polymorphic variants of APR[ex-4] and APR[Tc5b] were obtained, while APR[GLP1] and APR[GLP2] crystallized in a single form only (Supplementary Fig. 20, Supplementary Table 1).

Both APR[gluc] and APR[GIP] formed gel, or ultrathin needle-shaped microcrystals indicating aggregation propensity, but these were unsuitable for structure determination by X-ray crystallography. Crystals of APR[GLP1] grew only at a low pH ($pH \approx 2$, $T = 37$ °C) where the N-terminal glutamic acid self-converted to a pyroglutamic acid (pE) via a spontaneous ring closure. To avoid this side reaction and study the effect of protection of the N-terminal, we synthesized and crystallized its N-terminally acetylated form, Ac-EFIAWL. This compound showed similar spectral properties to APR[GLP1] (Supplementary Fig. 7). With these multiple 3D structures in hand, we aimed to reveal which interactions or structural properties could play a role both in stabilizing and reversing amyloid formation.

Thus far amyloid reversibility has been only attributed to LARKS (Low-complexity Aromatic-Rich Kinked Segments) found in stress granule-associated hydrogel-forming proteins[31]. In LARKS, the polypeptide backbone is highly kinked which prevents tight interdigitation of the side chains along the β-sheet interface, leading to a decidedly less stable adhesion between the mating β-sheets than in tight steric zippers of pathogenic amyloids[32]. Here we present 9 amyloid crystal structures of 4 different reversibly amyloidogenic APRs that (Fig. 4) possess an extended backbone carrying the characteristic torsion angles of β-strands (Supplementary Fig. 21, Supplementary Table 2). Despite their highly homologous primary sequences, steric zippers formed by the currently studied hexapeptides show remarkably different crystal structures. Seven of the nine structures present an antiparallel (AP) backbone of class 7 or 8 topology, while 2 peptides, those of APR[GLP2] and polymorph D of APR[Tc5b], form class 1 or class 4 steric zippers both comprising of parallel (PA) β-sheets (Fig. 4, Table 1). Estimating the relative orientation of the β-strands within the amyloid

structure composed of the full-length hormones is not straightforward even based on our results - especially in view of the structural heterogeneity of the APR structures we found - nevertheless, their intrinsic predisposition to form amyloids is unquestionable. Hydrophobic and polar interactions both contribute to the stability of the presented amyloid-like crystal structures. In the structure of APR[GLP1] the hydrophobic residues (Phe, Ile, Ala, Leu) form a hydrophobic core, while Trp and pGlu participate in H-bonding with each other and water molecules (Fig. 4a). A similar interface can be found in the structure of its N-acetylated form. The presence of the acetyl group and Glu instead of pGlu resulted in a lateral shift between adjacent β-sheets, allowing Glu to form H-bonds with waters located near chain termini and Trp side chains from neighboring β-sheets (Fig. 4b). The fact that this structure also belongs to class 7 suggests that the hydrophobic residues are the main determinants of the topology of the nanostructure. In the structure of APR[GLP2] Ile3 and Trp5 create a hydrophobic interface with Asp1 located at the same side, forming H-bonds with water and chain termini. At the other interface Phe2 and Leu6 form hydrophobic interactions, while Asn4 forms inter- and intrachain H-bonds (Fig. 4c). Two polymorphic crystal forms could be grown from solutions of APR[ex-4] (Fig. 4d–e). In the case of polymorph A, side chains presented an unusually high degree of flexibility. Side chains of Phe2 and the partially disordered side chain of Trp5 assembled into an aromatic cluster near the solvent channel and chain termini. The hydrophobic core is formed by Leu1 and Leu6 while intrasheet H-bonds of Glu-Glu side chains stabilize the interface. In polymorph B of the same system, shifted interfaces could be found which are stabilized by different interactions (Fig. 4d). S3 is stabilized by the hydrophobic interactions of the Leu, Phe, and Ile side chains, plus the intersheet Glu–Glu side chain H-bonds. S1 is formed by aromatic residues Phe2 and Trp5, while S2 is stabilized by hydrophobic interactions between Leu and aromatic residues and Glu–Trp side chain H-bonds.

Four polymorphic crystal forms were grown from solutions of APR[Tc5b]. Interaction networks formed in crystals of polymorphs A and C

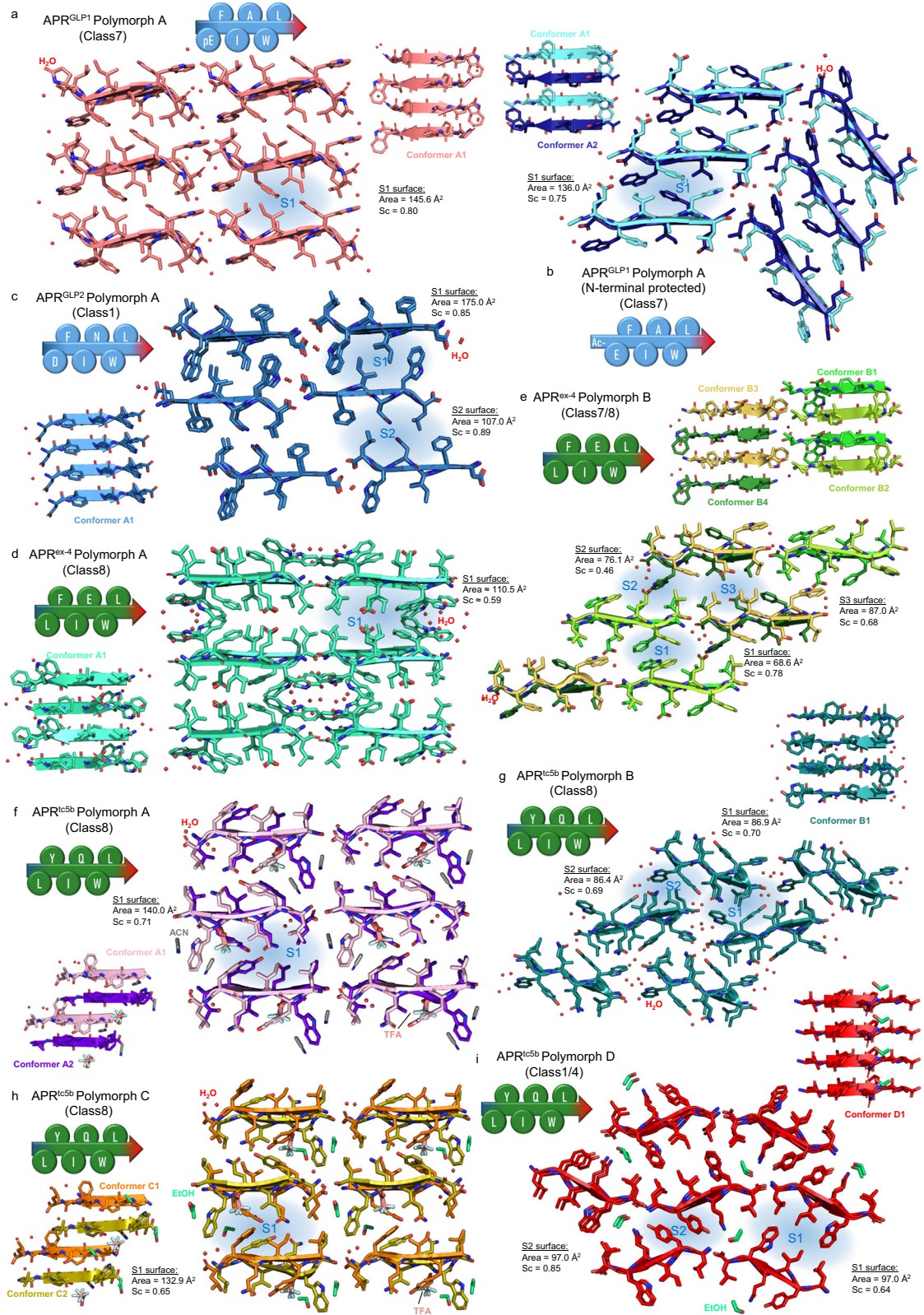

are essentially identical. Tyr2, Gln4, and Trp5 participate in several H-bonds with each other within the zipper interface. Aromatic side chains form a cluster in which acetonitrile or ethanol is also present, while TFA forms multiple H-bonds with termini and Trp side chains (Fig. 4f, h). Although the chains in polymorph B are also arranged according to the class 8 topology, different interactions stabilize the interfaces. While in polymorphs A and C Tyr, Gln, and Trp form

H-bonds at the β-sheet/β-sheet interface with each other, this arrangement allows their interactions with water molecules and chain termini (Fig. 4h). Surprisingly, polymorph D contains parallel β-strands (Fig. 3i). Ethanol is part of the crystal and forms H-bonds with the C-terminus and Gln4. Tyr and the N-terminus also participate in H-bonds, while Leu, Ile, Trp, and the ethyl group of ethanol form a hydrophobic core at the zipper interface (S2). This interface is similar to S1 in the

**Fig. 4 | Steric zipper morphology and β-sheet arrangements in amyloid-like APRs.** The interfaces of the hydrophobic zippers are indicated by the blue background and characterized by their respective contact area (Å$^2$) and shape complementarity value (Sc). **a** APR$^{GLP1}$ assembled in antiparallel, equifacial β-sheets with steric zipper belonging to the class 7 amyloid-topology, creating a similar interface to that found in case of (**b**) Ac-EFIAWL. **c** The class 1 amyloid topology of APR$^{GLP2}$ results in two different interfaces (S1, S2). **d** Polymorph A of APR$^{ex-4}$. Antiparallel, equifacial β-sheets formed steric zippers of class 8 topology. **e** Polymorph B of APR$^{ex-4}$. Antiparallel and equifacial β-sheets form a mixed topology of class 7 and class 8. A lateral shift between adjacent β-sheets allows the formation of 3 different interfaces (S1, S2, and S3). While S1 and S3 are class 7 interfaces, S2 is of class 8

topology. **f** Polymorph A and (**h**) polymorph C of APR$^{Tc5b}$: Crystals formed in both acetonitrile/water and ethanol/water mixtures of almost identical packing morphology with different solvent molecules (acetonitrile or ethanol) occupying the same spots in the crystals that belong to class 8. As pairs of antiparallel β-sheets shift along the fibril axis, 4 or 6 H-bonds form between the shifted chains. **g** Polymorph B of APR$^{Tc5b}$: Antiparallel β-sheets are assembled in class 8 topology. In contrast to all other crystal structures presented here, none of the laterally shifted neighboring β-sheets run parallel to each other, resulting in the formation of different interfaces. **i** Polymorph D of APR$^{Tc5b}$: This polymorph contains β-strands in the parallel arrangement of class 1 topology.

structure of DFINWL, as Trp5 and Ile3 form the core of this interface too.

The observed steric zipper interfaces are somewhat different from most of those that have been previously described in the literature. (Supplementary Table 3) Solvent molecules cluster near the backbone terminals and only dry (hydrophobic) interfaces can be observed. In the absence of solvent channels, several different contact interfaces (S1, S2, etc) are formed. Moreover, in contrast to the majority of other steric zippers formed by pathogenic amyloids that are characterized by tight interdigitation, in amyloids of APRs derived from secretin-like hormones the side chains make contacts only at their tips suggesting less adhesive intersheet contacts. This interaction mode may contribute to the reversibility of these amyloids. This structural feature is rooted in the outstanding proportion of large hydrophobic/aromatic residues forming the presented sequences. The bulky and branched side chains positioned at both sides of the β-strands make both surfaces hydrophobic, but their extensive size prevents the formation of close and compact, intertwined steric zippers. To compare the contact surfaces, we plotted the shape complementarity (Sc) values as the function of the areas (A$^2$) of the contacting surfaces. (Supplementary Fig. 22) The glucagon-like APRs (APR$^{GLP2}$ and APR$^{GLP1}$) show an extensive contact surface (Sc = -0.80; Area = ~160 Å$^2$) along the extended, linear backbone of high complementarity, while exendin-like sequences (APR$^{ex-4}$ and APR$^{Tc5b}$) present generally smaller values (Sc = -0.65; Area = ~110 Å$^2$).

Interestingly, exendin-derived hexapeptides showed the greatest structural plasticity that is reflected in multiple β-sheeted conformers giving rise to several polymorphs (Supplementary Table 2). Furthermore, in polymorph A of APR$^{ex-4}$ most side chains show an unusually high degree of flexibility, as Ile3, Glu4, and Trp5 all present at least two alternative conformers, while side chain atoms of both Leu1 and Leu6 exhibit unusually high B-factors. The weak residual electron density around the side chains of Trp5 suggests that additional conformers of this side chain are also present in the otherwise close-packed crystals.

In the presence of organic solvents, the exendin-like structures adopted more compact crystal forms (Fig. 4f, h, i), in which opposing side chains of the steric zippers interact with each other. Polymorphs formed in pure aqueous solution are laterally displaced, as the side chains of Tyr, Glu, and Gln rather form H-bonds with the surrounding water molecules (Fig. 4e, g). Additionally, a more complex level of packing polymorphism was observed in the case of amyloid-like crystal structures of APR$^{Tc5b}$. As the concentration of EtOH was reduced from 30% to 10% in the crystallization solution (keeping all the other parameters identical), the entire cross-β scaffold rearranged from parallel to antiparallel relative orientation (Fig. 4f, i). On the other hand, nearly identical polymorphic structures were obtained from the crystals grown in 10% EtOH and 30% ACN (polymorph A vs. C) (Fig. 4f, h). The two different organic solvent molecules are located approximately at the same place within the crystal, despite their different structures and chemical properties. Only two similar examples were published so far[33,34], however, APR$^{Tc5b}$ is the shortest sequence ever studied that forms polymorphic structures of both AP and PA arrangements. These

examples show that for some aggregation-prone regions several energetically quasi-equivalent amyloid states exist, and even subtle changes in the environmental conditions[35] could trigger the formation of fundamentally different polymorphic amyloid structures (Fig. 5).

We suggest that the palindromic nature of the primary sequence of exendin-derived APRs (Leu-aromatic-XX-aromatic-Leu) might be the basis of the observed polymorphism. Extended axial leucine-zippers can be formed along the β-spine at both terminals, regardless of whether the β-strands are oriented parallel or antiparallel (Fig. 6a, b). These Leu-zippers form an effective hydrophobic barrier between the dry-zipper and the solvent. In addition, they can greatly stabilize laterally shifted and hydrated β-ladders too.

## Different modes of neutralizing side chain charges promoting amyloid formation

Deprotonated Glu and Asp side chains in APRs act as gatekeepers[36] at neutral pH, as repulsive forces between negatively charged side chains and their hydration inhibit self-assembly and discourage dry steric zipper formation. Our study, which employed a combination of techniques, confirmed that Glu/Asp side chains of APR units are required to be neutralized (protonated) to allow amyloid formation.

The amyloid crystal structures of APR$^{GLP1}$, APR$^{GLP2}$, and APR$^{ex-4}$ revealed the nature of emerging interactions between charge-abolished side chains contributing to steric zipper stabilization. At pHs where Glu/Asp side chains become protonated, charge frustration ceases as side chains establish intersheet H-bonds. For example, protonated Glu forms H-bond either with the opposing Glu (S1 of polymorph A, S3 of Polymorph B in APR$^{ex-4}$ Fig. 6c), or with the indole rings of Trp(s) (S2 of Polymorph B Fig. 4e). These intersheet H-bonds are arranged similarly to those formed between Gln side chains in antiparallel polymorphs of APR$^{Tc5b}$ (Fig. 6d). The similar interaction modes of Glu/Gln residues suggest that if a negatively charged gatekeeper is neutralized, then its protonated form could be replaced by its amide counterpart. As a result of such mutation, the APR's amyloid formation becomes independent of the pH, as demonstrated by the enhanced amyloid formation propensity of APR$^{Tc5b}$ compared to that of APR$^{ex-4}$. Additionally, within a zipper, amide groups of Gln and Asn side chains can stabilize parallel β-sheets by axial H-bond ladders (Fig. 6e, f). In the crystal structure of pEFIAWL, the ring-closure of an N-terminal Glu residue (resulting in the formation of pyroglutamic acid) makes the N-terminal more hydrophobic by eliminating the charge frustrations linked to both the Glu side chain and the protonated N-terminal amine (Fig. 6g). The crystal structure of APR$^{GLP1}$ provide an example where pyroglutamic acid built in a β-spine amyloid nanostructure can be studied. Note that pyroglutamic acid formation of Aβ$_{E3-42}$ upon the cleavage of the N-terminal Asp-Ala dipeptide was proposed to account for the increased toxicity observed for the amyloid-β (Aβ$_{1-42}$)[37,38]. Toxicity is enhanced because the truncated amyloid-β, referred to as Aβ$_{pE3-42}$, forms oligomers faster and has an increased resistance against degradation compared to Aβ$_{1-42}$. Based on our current results, we propose that the elimination of the charges by ring closure, significantly contributes to self-assembly, compared to sole side chain

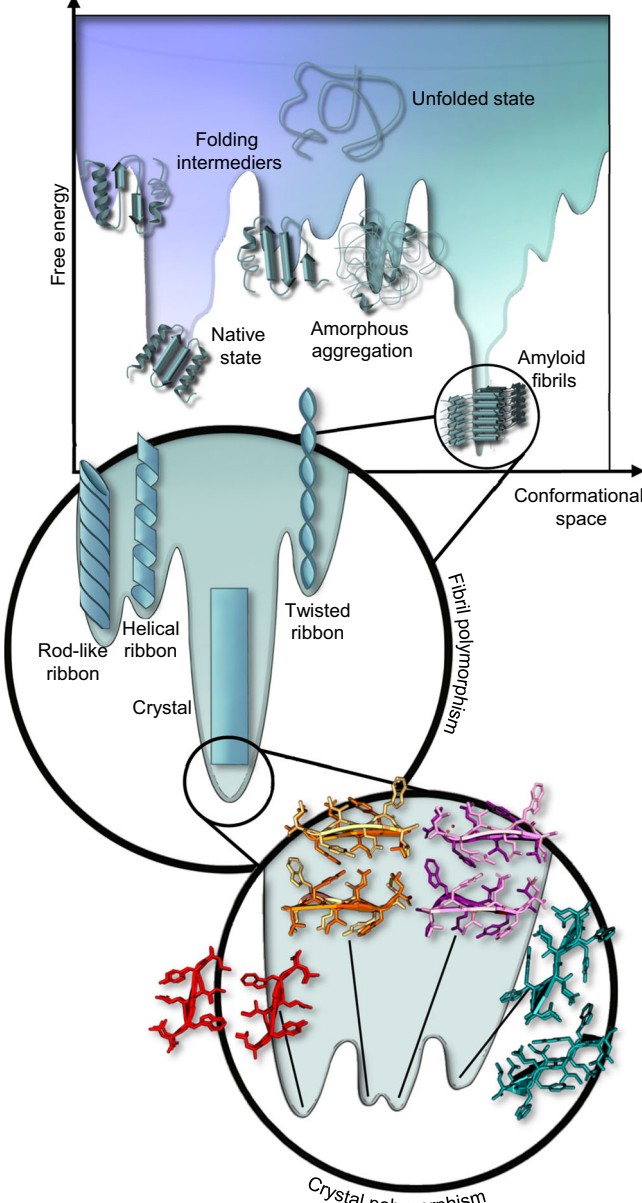

**Fig. 5 | Polypeptides and proteins folding (blue part) and aggregation (green part) combined energy landscape.** The low-energy folded state is separated from the amyloid states by kinetic energy barriers (transition states). In the currently proposed energy landscape[25] of fibrillar amyloids crystalline polymorphs occupy the lowest energy minima. Decidedly dissimilar amyloid structures can have similar relative energies, as found for APR^Tc5b. Therefore, a single aggregation prone sequence may adopt various fundamentally different amyloid topologies (e.g. both parallel and antiparallel). The exact parameters determining the fine structure of the complex amyloid forms remain to be pinpointed.

protonation of Glu. This might also be the reason why we observed the presence of pEFIAWL in the crystal state while the parent hexapeptide, EFIAWL, did not crystallize despite forming amyloid (Supplementary Fig. 14). The five-membered ring structure of pGlu shows similar geometry to that of Pro, which is considered a formidable β-sheet breaker. Thus, cyclic pGlu with quasi-fixed dihedrals would be expected to function as a gatekeeper of amyloid formation by twisting the backbone structures away from that of typical β-strands. However, the example of the crystal structure of pEFIAWL provides compelling evidence that a Pro-like residue right next to an APR does not necessarily prevent β-stranded amyloid formation.

## Amyloid nucleation and steric zipper formation via aromatic residues

The evolutionarily conserved Phe/Tyr and Trp residues of gastrointestinal hormones play a key role in receptor binding[16]. Here we draw attention to their importance in amyloid formation too, as their abundance is relatively high in APRs. Among the 88 previously published hexapeptides of known amyloid-like 3D structures, both Phe (28) and Tyr (27) are relatively frequent, but Trp only appears two times. The nine currently determined crystal structures all have two aromatic residues, one of which is Trp in all cases.

The comprehensive CD-based monitoring of the amyloid formation of APR^Tc5b in water unexpectedly revealed a distinct transient signal (T) dominating the initial amyloid nucleation (Supplementary Fig. 9: dark yellow curves at 15 min between pH range of 3.9–7.6). The nascent B-type CD curve is complemented by a highly intense negative/positive exciton couplet (at the wavelength of 223 nm /233 nm) indicating that a chirally perturbed π-π interaction plays a dominant role in the early phase of the self-association. This exciton couplet vanishes as β-strands of the emerging amyloid mature. Its temporary presence indicates that some intermolecular aromatic clusters (e.g. Trp↔Trp; Trp↔Tyr; Tyr↔Tyr) could have a role in amyloid preformation[39]. As no such transient T-type CD signal was observed for the same chromophores either at strongly acidic, or basic conditions, we propose that both the π-π interaction and the self-association triggered β-strand formation are pH-controlled. The above-described CD spectral shift (U → (T) → B-type) is detected exclusively for APR^Tc5b, but not for APRs containing Phe, indicating its specificity toward Tyr-Trp couples.

In line with the above, π-π interactions could be the thrust of early-stage amyloid formation and may have a major contribution towards β-spine stabilization too. Therefore, we have carefully analyzed the aromatic side chain interactions of all the steric zippers studied here. As the two aromatic side chains are located at the opposite faces of the β-strands, they can only form intermolecular aromatic interactions. They can either connect adjacent β-strands of the same β-sheet (intrasheet interaction), or different β-sheets via the dry zipper interface (intersheet interaction) (Table 2). Aromatic side chains form intrasheet aromatic ladders in the two parallel structures of APR^GLP2 and APR^Tc5b (polymorph D) (Fig. 6h, i), as well as in both polymorphic structures of APR^ex-4. However, intrasheet aromatic interactions are not present in crystal structures of the pEFIAWL and Ac-EFIAWL sequences, nor in the two antiparallel LYIQWL structures, since the Tyr/Phe and Trp side chains face in the opposite directions (Fig. 6j).

Intersheet aromatic interactions have been observed in the amyloid crystals of APR^GLP1 (Ac-EFIAWL), APR^ex-4, and APR^Tc5b. In the crystal structure of Ac-EFIAWL, the interaction between Phe residues from neighboring sheets is made possible by a slight shift of the sheets compared to the structure of pEFIAWL. An aromatic cluster is formed in both polymorphs of APR^ex-4. However, the exact geometry of the cluster in polymorph A is ambiguous due to the elevated internal mobility of the Trp side chain (Fig. 6k). In polymorphs A and C of APR^Tc5b, an acetonitrile/ethanol solvent molecule is incorporated in the aromatic cluster, leading to somewhat larger ring-to-ring distances as compared to those of APR^ex-4.

Strikingly, in AP β-sheets, the Trp side chains show the highest conformational heterogeneity. We assume that the bulkiness of Trp disfavors the formation of too compact amyloid-like crystals, while the presence of Trp is essential for receptor recognition and binding, making Trp a Janus-face residue. The multiple side chain orientations of Trp within the amyloid-like crystal form are the result of imperfect molecular packing, somewhat destabilizing the Trp contacting amyloids and thus contributing to amyloid reversibility. The hormone-derived APR segments studied here proved to be generally less

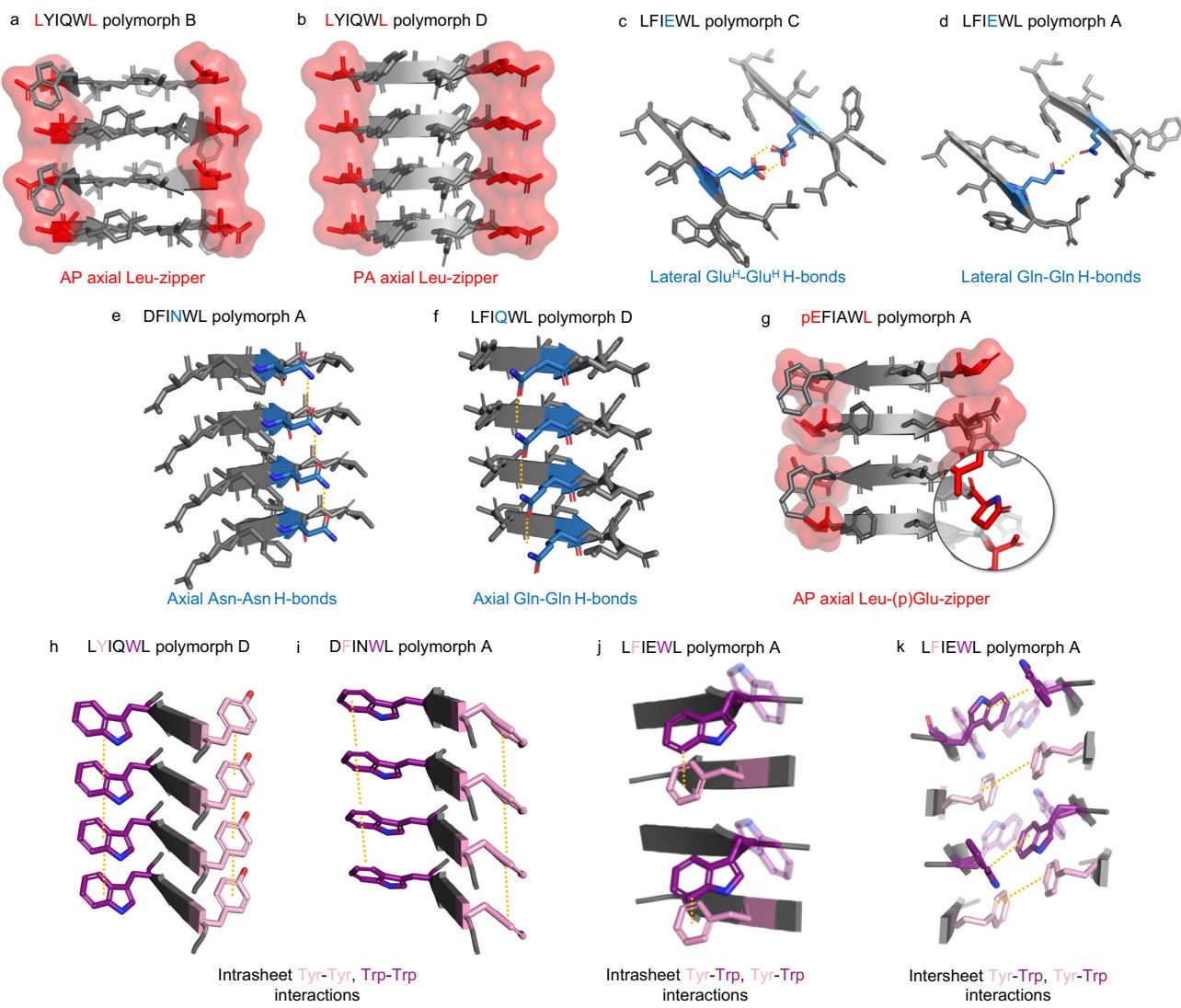

**a** LYIQWL polymorph B
AP axial Leu-zipper

**b** LYIQWL polymorph D
PA axial Leu-zipper

**c** LFIEWL polymorph C
Lateral Glu^H-Glu^H H-bonds

**d** LFIEWL polymorph A
Lateral Gln-Gln H-bonds

**e** DFINWL polymorph A
Axial Asn-Asn H-bonds

**f** LFIQWL polymorph D
Axial Gln-Gln H-bonds

**g** pEFIAWL polymorph A
AP axial Leu-(p)Glu-zipper

**h** LYIQWL polymorph D
Intrasheet Tyr-Tyr, Trp-Trp interactions

**i** DFINWL polymorph A
Intrasheet Tyr-Tyr, Trp-Trp interactions

**j** LFIEWL polymorph A
Intrasheet Tyr-Trp, Tyr-Trp interactions

**k** LFIEWL polymorph A
Intersheet Tyr-Trp, Tyr-Trp interactions

**Fig. 6 | Stabilizing interactions observed in amyloid crystal structures.**
Hydrophobic Leu-zippers are formed along the terminals of β-sheets in case of antiparallel (**a**) as well as parallel (**b**) directed β-strands of APR^Tc5b. **c** Protonated side chains of Glu stabilize the zipper contacts of APR^ex-4 by H-bonds. The strength of such an H-bond pair is well established by studies showing for example the elevated boiling point of glacial acetic acid. Gln side chains (**d**) of APR^Tc5b adopt a similar arrangement to those of Glu-pair above. Side chains of (**e**) Asn and (**f**) Gln stabilize

parallel β-sheets of APR^GLP2 and APR^Tc5b by H-bonds along the axis. **g** With the ring-closing reaction of *N*-terminal Glu a hydrophobic Leu-(p)Glu zipper is formed, that stabilizes β-sheets of APR^GLP1 along the axial direction. Typical intrasheet aromatic ladders (**h**, **i**) and intrasheet (**j**) or intersheet (**k**) aromatic clusters are observed. Tryptophan of LFIEWL (polymorph A) adopts different conformational states, (opaque and transparent side chains) having multiple geometries of aromatic interactions.

compactly packed in their amyloid nanostructures, making the pH-controlled disintegration and the release of the soluble and physiologically active monomeric form easier during excretion.

## Discussion
Gastrointestinal hormones must meet two essential requirements: they must sustain their biological activity and manage to be stored in large quantities without perturbing the cellular homeostasis. The herein studied hydrophobic, evolutionary conserved xFxxWL hexapeptide APRs of these hormones comply with both criteria. Our results provide evidence that the gatekeeper Asp and Glu residues within these APRs function as pH-controlled molecular switches. The emergence of the amyloid or unstructured states is dependent on the extent of protonation of these residues. pH-sensitive amyloid reversibility was suggested for glucagon[12], β-endorphin[14] and PMEL[40]. The herein investigated APRs of the glucagon-family compile these examples indicating a common mechanism for pH dependence which

allows for the storage of hormone peptides as amyloids in acidic granules. This concept is further supported by the example of non-natural APR^Tc5b, which—lacking a charged gatekeeper residue—remains in its amyloid state irreversibly over a wide pH range.

High resolution structural data concerning the amyloid form of gastrointestinal hormones is so far only available for glucagon. As it is generally believed that short APR segments of proteins initiate their amyloid formation[41,42], the presented steric zippers of incretin APRs (GLP-1, GLP-2, and exendin-4) might provide valuable insight into amyloid formation of these hormones too. Although crystal structures of APR hexapeptide segments necessarily differ from fibril structures of their full-length counterparts, their analysis and comparison with the reported fibril structures might point toward general features of their amyloid assemblies (Supplementary Figs. 23–24). First, it is interesting to note that the two experimentally determined structures of full-length glucagon fibrils exhibit different topologies, with parallel and antiparallel assembly depending on the experimental conditions,

**Table 2 | Observed aromatic interaction in the steric zippers**

| APR | Polymorph | Strand arrangement | Involved residues | Geometry | Ring-to-Ring distances (Å) |
|---|---|---|---|---|---|
| **Intrasheet aromatic interactions** | | | | | |
| GLP-2 | A | PA | Trp-Trp (S1 surface) and Phe-Phe (S2 surface) | face-to-face | ~4.9 |
| Tc5b | D | PA | Trp-Trp (S1 surface) and Tyr-Tyr (S2 surface) | face-to-face | ~4.9 |
| exendin-4 | A | AP | Phe-Trp (S1 surface) | displaced face-to-face | ~4.2 |
| | B | AP | Phe-Trp (S1 surface) | displaced face-to-face | ~4.6 |
| **Intersheet aromatic interactions** | | | | | |
| GLP-1 (Ac-EFIAWL) | A | AP | Phe-Phe (S1 surface) | displaced face-to-face | ~5.0–6.0 |
| exendin-4 | A | AP | multiple Phe and Trp (S1 surface) | aromatic-cluster | ~4.5–6.0 |
| | B | AP | multiple Phe and Trp (S1 surface) | aromatic-cluster | ~4.5–6.0 |
| Tc5b | A&C | AP | multiple Phe and Trp (S1 surface) | aromatic-cluster | ~5.1–7.0 |

supporting the hypothesis that secretin-like hormone peptides may exhibit polymorphism[43] in their amyloid forms. All presently determined APR models adopt an extended β-stranded backbone conformation and maintain some degree of structural plasticity even within their amyloid nanostructures by harboring multiple side chain conformations. The orientation of the β-strands tends towards antiparallel upbuild, however as polymorphic amyloid structures of APR^Tc5b also show, both parallel and antiparallel orientation can be adopted by the same primary sequence. Thus, AP *vs.* PA β-sheet formation is regulated not simply by the primary sequence of the polypeptide chain, but also by additional factors. None of the dry-interfaces exhibit a tight interdigitation of the side chains indicating less adhesive contact between the mating β-sheets. The APRs presented here, as well as full-length glucagon fibrils, are capable of forming multiple distinct dry steric-zipper interfaces. In the AP-oriented fibrils formed by glucagon, two types of conformers self-associate in the alternating planes (Supplementary Fig. 12a). The odd-numbered residues of conformer A form a steric zipper interface (Supplementary Fig. 12c), whereas even-numbered residues of conformer B line up, as a different steric zipper interface (Supplementary Fig. 12e), similarly to that seen for APR crystals in class 8 (Supplementary Fig. 12b, d, f). Two different (S1, S2) possible contact surfaces were also found in APR crystals belonging to the class 1 antifacial family. (Fig. 4c, i) The conformational plasticity and the pH-dependent nature of amyloid formation together provide the basis of reversibility.

The observed disassembly of APR fibrils (Supplementary Fig. 10–15) and that of full-length glucagon (Supplementary Fig. 18) upon a change in pH from acidic to neutral suggests that the same "switch-mechanism" may be active in both cases. However, according to our results, the rate of disintegration is much faster in the case of the aggregates of short APR hexapeptides, suggesting a less complex fibrillar structure and the more solvent-exposed position of the gatekeepers. In parallel-oriented glucagon fibrils, the acidic gatekeepers located at Asp21 are exclusively found on the surface completely exposed to the solvent and thus do not contribute to the lateral destabilization of the central steric zipper upon deprotonation. However, when the side chain of Asp21 becomes negatively charged, the parallel β-sheet is destabilized vertically along the fibril axis (Supplementary Fig. 24b). In contrast, some protonated Asp gatekeepers are buried in the dry-zipper of the antiparallel glucagon fibril, although they are not entirely isolated from the solvent. Cavities near Asp9 and Asp21 in conformer A create solvent-accessible channels that may facilitate the deprotonation of the gatekeepers, which could then lead to the lateral destabilization of the nanostructure. (Supplementary Fig. 23c–e).

The aggregation propensity of globular proteins is generally matched with their cellular concentration[44], consistent with protein solubility[45], slightly exceeding their physiological expression levels.

Amyloidogenic sequences, buried inside the hydrophobic core of globular proteins are deeply conserved and considered as a secondary and negative side effect originating from the specific requirements of protein thermodynamic stability[46]. Contrary to globular proteins, APRs of hormone peptides cannot be shielded by 3D-folding, as they are too small for the build-up of such complex structures. Moreover, their receptor binding surfaces and their APRs overlap, and thus they must remain exposed to solvent to preserve function. For this reason, tight control mechanisms evolved to uphold reversible amyloid formation of peptide hormones, while also allowing them to properly fulfill their physiological function. Secretory granules allow the long-term intracellular and isolated storage of aggregation-prone hormones at high concentrations in the form of amyloids at low pH. Enrichment of acidic gatekeepers (Glu/Asp) regulates the aggregation propensity of the APRs as a function of the pH. Furthermore, permanently charged basic gatekeepers (Arg/Lys) abundant at the flanks of the APRs increase the overall solubility of the polypeptides and control aggregation via charge repulsion. As soon as secretion into the bloodstream is triggered, amyloid-deposited hormones experiencing the pH shift coupled to the change in their surroundings disassemble to monomers and adopt their typically helical structures, effectively masking the amyloidogenic nature of their APRs. However, this modest refolding cannot fully diminish the aggregation propensity as the amphipathic helices are capable of forming coiled-coil dimers[47], which might well be the first step along pathways leading to oligomer formation[48]. Additional features have evolved to counteract aggregation, such as, for example, the short polyproline helix found at the *C*-terminal of exendin-4, which does not participate in receptor binding, but effectively shields the APR by forming a stable Trp-cage tertiary structure. Baets et al.[49] demonstrated that proteins of higher aggregation propensity tend to have a shorter lifespan. Thus, the efficient degradation of high-turnover proteins seems sufficient to preclude aggregation. In light of the above, it is not surprising that the lifetime of the secretin-like gastrointestinal peptides in circulation is only a couple of minutes due to fast digestion by DPP-IV[50]. This lowers the risk of hormone misfolding and amyloid formation, as homeostasis initiates elimination as soon as signal transduction gets completed.

Considering that these hormones spend most of their lifespan stored as amyloids, we propose that instead of their biologically active monomeric form, the original native fold of these molecules may actually be the energetically more stable amyloid structure (Fig. 5). The proglucagon-derived hormones investigated here have a common ancestral origin and their diverging functions are conserved in all vertebrates[17]. The dual function of the xFxxWL sequence seems to be unique and preserved during evolution (Supplementary Fig. 1) as reflected by the high degree of sequence conservation

between species. Through phylogenetic analysis and examination of their respective functions, the proglucagon ligand-receptor pairs are believed to have evolved independently before acquiring specificity for one another. This indicates that the typical receptor-ligand interactions and the specific function of hormones evolved much later than the common aggregation-prone xFxxWL motif appeared[16]. Additionally, some of these receptor genes have been lost in several different vertebrate lineages indicating changes in the physiological functions of these hormones[51]. For example, the receptor of GLP-1 is not present in the genome of bony fish, even though GLP-1 is present and promotes glucose production instead of acting as an incretin[52].

The amyloid aggregation potential of bioactive peptides is a double-edged sword. The aggregation potential of polypeptide-based medications[53] is a threat to avoid in the pharma industry. On the other hand, a controlled and reversible self-assembly of hormone derivatives into nanofibrillar amyloid depots promises new perspectives in the development of long-acting drug delivery approaches[54,55]. Revealing the aggregation propensity of proglucagon-derived peptides gains special significance in light of the fact that more than 50 such peptides are already in clinical trials or on the market[20].

## Methods

### APR hexapeptides synthesis

APR hexapeptides were prepared by using our in-house developed flow chemistry-based solid-phase peptide synthesizer, using the Fmoc/tBu strategy[56,57]. Preloaded Fmoc-Leu-Wang TentaGel resin (containing the first C-terminal amino acid) was used. Coupling was performed with OxymaPure and DIC reagents, while DMF was used as the solvent. The reaction took place at 80 °C under a pressure of 7−9 MPa. Oligopeptides were cleaved from the resin by using the mixture of 2.5 v/v% triisopropylsilane, 2.5 v/v% water and 95 v/v% TFA, at room temperature, with continuous stirring for 3 h. TFA was removed by rotary vacuum evaporator and the oligopeptides were precipitated in cold diethyl ether. After sedimentation, the ether was decanted, and the sediment was washed again with fresh ether. This cycle was repeated three times, followed by vacuum drying. The raw oligopeptides were dissolved in 5:95 v/v% of ACN:$H_2O$ and filtrated (PTFE membrane: 45 μm). The oligopeptide solutions were purified on a reverse-phase HPLC using a C12 column (Jasco LC-2000Plus HPLC system equipped with Jupiter® 10 μm Proteo 90 Å LC column 250 × 10 mm) and a gradient elution (ACN/water containing 0.1% of TFA). The collected samples were immediately frozen and lyophilized, aliquots were reserved for crystallization. Analytical purity (Supplementary Fig. 25) was presented both by MS (HR MS-Orbitrap) and analytical HPLC (Aeris™ 3.6 μm PEPTIDE XB-C18 LC Column 250 × 4.6 mm).

### APR hexapeptide samples for monitoring amyloid formation by far-UV CD

The lyophilized oligopeptides were dissolved in distilled water (concentration range of 0.15−0.20 mM: 0.13 mg ml$^{-1}$). The pH of each sample was adjusted using 0.01/0.1/1 M NaOH and HCl solutions (Orion Star A211 pH meter (Thermo Scientific™)). The samples were continuously stirred by a magnetic stirrer (500−600 rpm) and incubated (37 °C). The stock solutions without any dilution were directly measured at given times (0−168 h): both concentrations and chiroptical properties were determined by NanoDrop Lite Spectrophotometer (Thermo Scientific™, at 280 nm, using the proper $\varepsilon_{Y/W} = 6990$ M$^{-1}$cm$^{-1}$, $\varepsilon_{F/W} = 5500$ M$^{-1}$cm$^{-1}$ molar extinction coefficients) and CD-spectroscopy.

### Glucagon fibril preparation

Glucagon was obtained from Novo Nordisk (Bagsvaerd, Denmark) as GlucaGen®HypoKit® single-dose injection kit. The lyophilized glucagon powder was dissolved in water, which was provided in the kit, for reconstitution. The resulting stock solution contained glucagon at a concentration of 1 mg ml$^{-1}$, along with lactose monohydrate. The pH of the stock solution was measured as 3.3, consistent with the kit description. Subsequently, the glucagon stock solution was incubated for a day at 37 °C and stirred at a rate of 500−600 rpm using a magnetic stirrer. Aliquots were taken and directly transferred to freshly cleaved mica or measured by CD after a twofold dilution.

### Far-UV CD measurements and data processing

The FUV-CD measurements were performed on JASCO (Tokyo, Japan) J-810 and J-1500 spectropolarimeters. The temperature of the cells was controlled by a Peltier-type heating system. JASCO Spectra Manager v2.0 was used to collect and process data. Samples were measured in 1.0 mm path-length quartz cuvettes at 25 °C. Each spectrum was the average of three scans collected as follows: spectral scanning speed of 50 nm/min; the bandwidth of 1 nm; 0.2 nm step resolution over the wavelength range of 185−260 nm (far-UV). All spectra were corrected by subtracting the solvent spectrum and by smoothing with a convolution width of seven, using the Savitzky-Golay method. The raw ellipticity (mdeg) data were converted into mean residue molar ellipticity units ($[\theta]_{MR}$/ deg cm$^2$ dmol$^{-1}$).

### Sample preparation for FTIR and AFM-based analysis of amyloid formation

APR peptide solutions with a final concentration of 6 mM were prepared and adjusted to pH 7.2 and these were used for recording the initial FTIR spectra. Afterward, the pH of each sample was adjusted to 4.2 and FTIR spectra were recorded again immediately. At each sampling point, 30 μl aliquot was transferred to the ATR cavity of the IR equipment, while a 5 μl of concentrated and 5 μl of 10 times diluted (with properly pH adjusted water) aliquots were spread onto freshly cleaved mica surface and dried overnight in a vacuum-exicator. The pH of the amyloidogenic IR samples (at $pH = 4.2$) that had matured for 24 h (by stirring and incubating at 37 °C) was then reset to pH 7.2 prior to the reversibility measurements.

### Fourier-transform infrared spectroscopy (FTIR) measurements

FTIR measurements were performed using a Bruker (Billerica, MA, USA) Equinox 55 FTIR spectrometer equipped with a bio-ATR (attenuated total reflectance) cell, where the internal reflection element is made of a ZnSe crystal. The ZnSe photoelastic modulator of the instrument was set to 1600 cm$^{-1}$, and an optical filter with a transmission range of 1900−1200 cm$^{-1}$ was used to increase the sensitivity in the amide I−II spectral region. The MCT (mercury-cadmium-telluride) detector was cooled with liquid nitrogen. Each FTIR spectrum was recorded by averaging 128 scans in the range of 4000 cm$^{-1}$ to 850 cm$^{-1}$ with a resolution of 4 cm$^{-1}$ using an aperture of 3000 microns. The IR spectra were calculated from the single-channel DC spectra, and baseline correction was applied to each measurement by subtracting the blank solvent spectrum. Software OPUS 6.5 were applied for data procession and exporting.

### Atomic force microscopy (AFM)

The surface morphology was analyzed using a FlexAFM microscope system (Nanosurf AG, Liestal, Switzerland), operating in dynamic mode controlled by Nanosurf control software C3000 version 3.10.4. The measurements were taken using Tap150GD-G cantilevers (BudgetSensorsLtd., Sofia, Bulgaria), with a nominal tip radius of less than 10 nm. Low-resolution pre-screenings were conducted before data collection to prevent significant height variations in surface topology and identify fibril-like formations. Data collection was performed once from various locations within a single sample. Images were captured in close proximity to optically dense aggregates on the surface, with a window size of 10 μm × 10 μm and a resolution of 512 pixels/line. In several cases, data were collected from the region of interest at a

higher resolution. Representative line profiles have been extracted from the images to characterize the cross-section of the filaments. Gwyddion 2.62 software was used to process the AFM data and generate the images.

## Thioflavin-T measurements

Stock solutions with a peptide concentration of 0.5 mM were prepared as follows: peptides were dissolved in water with pH 7.4 and sonicated for 10 min. The pH was then precisely adjusted to pH 7.4 and sonicated again. Centrifugation was performed for 2 min at 13000 rpm to ensure that only dissolved monomers remained. The exact concentrations of the supernatant were determined using NanoDrop. 185 µL of the peptide stock solution was added to black-walled 96-well microplates with flat bottoms (Greiner-Bio-One, Frickenhausen, Germany). ThT (Acros Organics - Thermofisher, Geel, Belgium) stock solutions (134 µM) were prepared by adjusting the pH of distilled water to pH 7 and 4 and filtering through a 0.44 µm filter. The exact concentration was determined by UV spectrometry (Jasco V-660 Spectrophotometer, Tokyo, Japan) at 412 nm with an extinction coefficient of 31.600 M$^{-1}$ cm$^{-1}$. The stock solution was stored in the dark at 4 °C. A twofold diluted ThT stock solution was added to the wells, resulting in a final ThT concentration of 5 µM. The pH of the peptide solution in half of the wells was adjusted to pH 4.0 before starting the ThT kinetics experiment. The plate was hermetically sealed to prevent evaporation. A SpectraMax iD3 microplate reader (Molecular Devices, Sunnyvale, CA, USA) controlled the experimental conditions (37 °C, orbital shaking at medium intensity) and collected fluorescence data over 24 h. The excitation and emission wavelengths were 445 and 490 nm, respectively. ThT fluorescence intensity was measured from the bottom of the microplate with medium photomultiplier tube sensitivity over an integration time of 400 ms. The fluorescence intensity of three parallel measurements was averaged and corrected by subtracting the background (pH-adjusted distilled water). The mean and standard deviation of ThT fluorescence intensities were plotted as a function of time. The ThT, CD and FTIR data were analyzed, processed, and plotted using Origin software version 2020b and Excel software versions 2016 to 2020.

## Determination of the theoretical p$K_a$ values and microspecies distribution

The p$K_a$ values of individual functional groups were predicted using PROPKA 3.5, which is available on the Github repository: https://github.com/jensengroup/propka. PROPKA 3.5 utilizes 3D structural data to make these predictions. By default, ionizable groups within three covalent bonds are penalized and excluded from the output. However, in order to access the p$K_a$ values of the N-terminal Asp residues, this default setting was switched off. The structural models were derived from the crystal structures of the APRs, with the exception of APR$^{gluc}$ and APR$^{GIP}$ for which we could not obtain atomic resolution data. APR models for these systems were derived from the amyloid fibril structures of glucagon, selecting two topologically different arrangements from the solid-state NMR ensemble of antiparallel glucagon amyloid (6NZN) based on the orientation of the Asp1 and Trp5 side chains, and the corresponding six residue segment of the cryo-EM structure of parallel oriented filaments (7XM8). GIP APRs were created from these models by implementing Q4N mutations. The new side chains were relaxed in 5000 step local Monte Carlo Multiple Minimum searches (Schrödinger Release 2023-2: MacroModel, Schrödinger, LLC, New York, NY, 2023.) involving only the side chain torsions of the Asn4 residues. The p$K_a$ values of the applied structural models were collected in Supplementary Table 4, and the averaged values were used for further analysis.

In order to plot the relative abundance of the charged APR microspecies as the function of the pH the calculated p$K_a$ values were converted to logK values respectively, and then they were applied in the following equations:

$$\alpha_i = \frac{\beta_i \times [H^+]^i}{1 + \sum_{i=1}^{n}(\beta_i \times [H^+]^i)}, \text{ and } \alpha^- = 1 - \sum_{i=1}^{n}\alpha_i$$

where, $\alpha_i$ is the mole fraction of $i^{th}$ macroscopically protonated species, $\alpha^-$ is the molar fraction of the fully deprotonated state, $\beta_i$ is the cumulative protonation constant of the $i^{th}$ species (the product of the stepwise $K_{1 \to i}$ protonation constants), and $[H^+]$ represents the hydrogen-ion concentration. The overall charge is defined as: $\bar{z} = z^- + \bar{n}$, where $z^-$ is the charge of the fully deprotonated state, and $\bar{n}$ is the average number of bound protons:

$$\bar{n} = \frac{\sum_{i=1}^{n}(i \times (\beta_i \times [H^+]^i))}{1 + \sum_{i=1}^{n}(\beta_i \times [H^+]^i)}$$

## Crystallization of the APR hexapeptides

LYIQWL, polymorph A: Initial crystals formed after incubating the oligopeptide solution instantly after HPLC purification in the eluent at 4 °C. The eluent contained ~67% H$_2$O, ~33% acetonitrile and ~0.1% TFA. Crystal formation could be reproduced by dissolving lyophilized peptide in 30% acetonitrile either with or without the addition of 0.1% TFAat a concentration of 0.2–1.0 mg ml$^{-1}$ and incubation at 4 °C, or 37 °C for several days. LYIQWL, polymorph B: Lyophilized oligopeptide was dissolved in water (c ~0.15 mg ml$^{-1}$) followed by pH adjustment with NaOH to pH 4.0–6.0: crystals grew at 37 °C. LYIQWL, polymorph C: Lyophilized oligopeptide was dissolved in 10 v/v% ethanol at 0.5–1.0 mg ml$^{-1}$ concentration and incubated at 20 °C overnight. LYIQWL, polymorph D: Lyophilized oligopeptide was dissolved in 20-30 v/v% ethanolat 0.5–1.0 mg ml$^{-1}$ concentration and incubated at 20 °C, or 37 °C overnight. DFINWL: Lyophilized oligopeptide was dissolved (c ~0.6 mg ml$^{-1}$) in 30% acetonitrile 70% H$_2$O and 0.1% TFA, at 37 °C and incubated for 4 weeks. pEFIAWL: Lyophilized oligopeptide EFIAWL was dissolved (c ~0.15−0.5 mg ml$^{-1}$) in 30% acetonitrile, 70% H$_2$O and 0.1% TFA at 37 °Cand incubated for several weeks. According to MS analysis, samples used for CD measurements did not contain pEFIAWL, while both EFIAWL and pEFIAWL were present in the crystallization samples. Note that the incubation period for the CD measurements was a week-long only, while crystallization took over four weeks, moreover, the CD sample was absent of acetonitrile too. Ac-EFIAWL: Lyophilized oligopeptide was dissolved in 0.1 M citrate buffer, pH 4.0 (c ~0.016 mg ml$^{-1}$) and incubated at 37 °C. LFIEWL, polymorph A: Lyophilized oligopeptide was dissolved (c ~0.15−0.5 mg ml$^{-1}$) in 30% acetonitrile, 70% H$_2$O, and 0.1% TFA at 37 °C and incubated for several weeks. LFIEWL, polymorph B: Lyophilized oligopeptide was dissolved in water (c ~0.15 mg ml$^{-1}$), followed by a pH adjustment with NaOH to pH 5.50. Crystals appeared after incubation at 37 °C for several days.

## X-Ray diffraction measurements and structure determination

Single crystal X-ray diffraction data were collected at 100 K on a Rigaku (Tokyo, Japan) XtaLab Synergy-R rotating anode diffractometer, by using Cu Kα radiation. Data collection and reduction were carried out using CrysAlisPro.

The phase problem was solved by molecular replacement, using idealized 5−6 residue long β-sheets or polyalanine chains created from our previously solved amyloid structures as search models, either in Phaser of the Phenix package[58], or by using Fragon[59] in the CCP4 package[60] when data of sufficient resolution were available. A manual model building was carried out in COOT[61], then models were refined using Phenix.refine[62].

In polymorph A of LFIEWL, the disorder of the side chain of Trp5 made model building complicated. Two possible rotamers of Trp5 corresponding to the strongest electron density peaks were built into the model, but residual electron density suggests that more than 2 rotamers are present. The strongest residual electron density peaks were modeled as water molecules. The high extent of its disorder explains the unusually low quality of the electron density map around Trp5.

In DFINWL disordered solvents are present in the crystals, forming channels that we could not model with single solvent molecules. TFA, acetonitrile, and water molecules were tried, but the addition of further solvent molecules resulted in either higher R factors, or molecules that did not fit into the electron density. For this reason, these regions of the asymmetric unit remained unmodeled.

Figures for atomic models were prepared using PyMOL software version 2.5.5.

### Description of the steric zippers

Sc[63] from the CCP4 suite was used to calculate shape complementarity. Areaimol[64], also from the CCP4 suite, was used to calculate the buried surface area. The first sheet pairs were generated with PyMol: 3 + 3 strands for parallel and 6 + 6 strands for antiparallel (i.e. 3 times the repeat unit of a sheet). The shape complementarity was calculated for the whole generated pair of sheets. The buried area was calculated as the sum of the solvent-accessible area of the central strands buried in the sheet minus the solvent-accessible area of the central strand pairs and averaged over the number of strands in the repeat unit of the sheet pair.

### Reporting summary

Further information on research design is available in the Nature Portfolio Reporting Summary linked to this article.

## Data availability

The processed CD, FTIR, and ThT data generated in this study are provided in the Source Data file. Raw data of the FTIR and CD spectra, as well as the unprocessed AFM pictures, will be shared upon request by contacting the corresponding author. Novel structural coordinates have been deposited online in the Protein Data Bank under accession codes: 8ANJ; 8ANK; 8ONQ; 8ANN; 8ANL; 8ANH; 8ANM; 8ANI; 8ANG. Polymorph glucagon structures of 6NZN and 7XM8 were used in the discussion of this study. Source data are provided with this paper.

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

## Acknowledgements

We would like to express our gratitude to Veronika Harmat, Eszter Szabó and Zsolt Fazekas for their valuable insights during discussions. We also acknowledge the contributions of Balázs Boller for collecting the selected data and Boglárka Schilling-Tóth for assisting with ThT measurements. This work was completed under the ELTE Thematic Excellence Programme (D.H., Zs.D., F.B., G.Gy., N.T., M.S.E., D.K.M., A.P.), which received support from the Hungarian Ministry for Innovation and Technology. The research was conducted as part of two projects, namely No. VEKOP-2.3.2-16-2017-00014 and VEKOP-2.3.3-15-2017-00018 (D.H., Zs.D., F.B., N.T., M.S.E., D.K.M., A.P.), which were co-funded by the European Union, the State of Hungary, and the European Regional Development Fund. Additionally, this work was supported by project no. 2018-1.2.1-NKP-2018-00005 (D.H., Zs.D., F.B., G.Gy., N.T., M.S.E., D.K.M., A.P.) and project no. NKFIH FK 142754 (G.Gy), both of which were implemented with funding provided by the Hungarian National Research, Development and Innovation Office. Finally, project number RRF-2.3.1-21-2022-00015 (D.H., Zs.D., F.B., G.Gy., N.T., M.S.E., D.K.M., A.P.) was implemented with the support of the European Union's Recovery and Resilience Instrument.

## Author contributions

D.H. and A.P. designed and coordinated the project. D.H. synthesized and purified the peptides, and with Zs.D., N.T., and F.B., prepared and crystallized the samples. N.T. and D.H. conducted CD and ThT measurements, D.H. and F.B. measured the FTIR spectra, and D.H. and G.Gy collected the AFM recordings. Zs.D. measured and processed the crystal structures, while Zs.D. and M.S.E. analyzed the structural data. D.H. processed the CD, ThT, AFM, and FTIR data. P.H. determined the distribution of the microspecies. The manuscript was written by A.P., D.H., Zs.D., D.K.M., and all authors contributed to it. A.P. provided the funding and extensive technical and instrumental background over the years.

## Funding

## Competing interests

The authors declare no competing interests.
