## [Peer Review File · Nature Communications]

REVIEWER COMMENTS

Reviewer #1 (Remarks to the Author):

Perczel and coworkers present an interesting study of a number of hexapeptides with the conserved glucagon family motif xFxxWL. The peptides show a tendency to beta-sheet structure which roughly follow the population of the NH₂-peptide-COO⁻ species. The authors also provide crystallographically based structures of the hexapeptides which highlight different isoforms with varying contact interfaces and with structural clues to the observed reversibility of beta-sheet formation. I have a number of issues to address before I can recommend publication:

Major issues:

1. One major limitation is the technique used to measure amyloid formation. The authors claim that “The aggregation propensities of hexapeptides derived from the gastrointestinal hormones were monitored by Circular Dichroism (CD) spectroscopy in the far-UV range.” No, CD monitors secondary structure changes and at best indirectly follows aggregation. And they do not provide any direct evidence for claims such as “the solubility of the hexapeptides is enhanced to such an extent that the equilibrium gets shifted toward the disassembled and solvated monomeric forms” – there are no solubility data. Aggregation can be measured by determining the solubility of the peptides (e.g. absorption before and after centrifugation to remove insoluble species) while amyloid formation can (to some approximation) be determined with ThT binding (though it is not quantitative in the case of e.g. polymorphism and pH changes can also affect it). The authors do not provide any direct evidence for amyloid formation under any conditions and this must be addressed carefully and thoroughly, not necessarily in all the data in Ext. Fig. 2-7 but certainly to a representative extent using methods above or equivalent alternatives.

2. While I think the authors have a good case regarding pH-driven aggregation (subject to the extensions described above), the data are currently based on rather qualitative comparison of a large group of different CD spectra. It would improve the message considerably if the authors could encapsulate the observed changes in terms of extent of beta-sheet formation, e.g. by deconvoluting the spectra using the basis spectra in Fig. 2A and making a sensible estimate of amyloid (beta-sheet) formation as a function of pH from the many pH-varying spectra in Ext. Fig. 2-7. This could then be plotted in panels A of Ext. Fig. 2-7 plus Fig. 2CD.

3. All the data above are from hexapeptides rather than the peptide hormones themselves which are the important biological species. How reversible is their amyloid formation with pH changes? Can the authors address this experimentally?

Minor issues:

1. Fig. 2B: it is unclear what the state “T” refers to. The authors refer to a “transition structure” but this is a vague term. The spectra referred to as T in Ext. Fig. 7D-H (orange brown) should first of all be clarified as being 15-min spectra and secondly it seems that they are combinations of the beta and beta’ spectra shown in Fig. 2A – so could they not just be a mixture of these two states?

2. pH reversibility (ext. figures 2-6 panel J): It would help the reader if the authors also showed the pH 7.4 spectrum of each peptide when dissolved directly at pH 7.4 (i.e. without the amyloid "detour").
3. I think the authors have a very good case for reversibility of peptide structure, but it would be good if they could show complete regain of solubility when reverting to pH 7.4, e.g. by comparing spectra before and after centrifugation.
4. "polymorph" => "polymorph" in several of the figures (e.g. Fig. 5); similar for "amiloid" in the text

Reviewer #2 (Remarks to the Author):

The work presented by Horváth et al. deals with the amyloid properties and their dependence on the solution pH of short hexapeptides, sharing an xFxxWL motif recurrently found in peptides of the glucagon family. They use circular dichroism to monitor the transition between secondary structures as a function of the pH and peptide crystallization and X-ray diffraction to gather molecular details of the different assemblies.

Major concerns:

1) Although the authors devoted significant effort to solving eight different crystal structures of peptides belonging to the same or different hormones, and their models are interesting from a chemical point of view and can provide clues for designing reversible assemblies, I am not convinced that these short peptides provide new relevant information of biological interest in a context in which the amyloid structure of full-length amyloid hormones, as IAPP or glucagon itself (<https://doi.org/10.1101/2022.11.21.517306>) have been solved by Cryo-EM, in this later case in acidic conditions, where the assembly is permitted. Indeed, even if it is true that the DFVQWL segment of glucagon is involved in the central steric zipper that sustains the inner part of glucagon fibril, the contacts it establishes are different from the equivalents ones shown in the crystal states of the different peptides, essentially because, in the Cryo-EM structure, residues out of this region interact with this segment providing additional stability to the axis of the fibril. In addition, an important role is played by two positive residues adjacent to the segment studied here. Indeed, the presence of cationic residues adjacent to the hexapeptide seems to be a conserved feature of the family, and its inclusion in the studied peptides would have been relevant.

2) From the sentence, "This reversibility is triggered by the complete protonation of the C-terminal carboxylate of LYIQWL, yielding a monopole state with a positively charged N-terminal. In contrast to the wild type sequences discussed above. " and the protonation graphs, it is deduced that in the used peptides, the N-terminal and C-terminal ends are not protected, and thus, the peptides bear two

additional charges that respond to the pH but that are not present in the context of the natural sequence. This is surprising, since it is a common practice in the amyloid field to protect such ends to discard their potential contribution to the conformational conversion, especially when the net charge of the sequence is relevant for the experiments, as this is the case. This, together with the fact that the conditions used in crystallography are clearly far away from physiological ones, either extracellular or intra-secretory granules, including the use of 30% of organic solvent and pHs as low as 2-3 using TFA, argue against the observed structures and interactions occurring in a biological context.

3) Unexpectedly, the amyloid nature of the structures at the different pHs used to monitor secondary structure using CD is not demonstrated. Techniques like Thioflavin-T and Congo Red binding/staining (or alternative amyloid-dyes), light scattering, and Transmission Electron Microscopy are conventional in the field and should have been used. In their absence, the formation of an inter-molecular β -sheet of amyloid nature at around neutral pH is arguable.

4) The protonation state of the different pHs is calculated theoretically from the pKa of the individual residues in the sequence. However, it would have been more correct to calculate it experimentally in the different solutions at the peptide concentrations used in the CD experiments, for example, by measuring the sample's Z-potential.

5) The assignment of the different conformational states, unfolded and β -sheet, by CD is not convincing since most spectra are not canonical, not displaying the typical signatures of this type of secondary polypeptide conformations. This is normal for such short peptides with high aromatic content strongly influencing the UV-spectrum. Moreover, CD is not the best technique for measuring peptides of low solubilities, such as those in this study, because it is only sensible to the peptide in solution. Accordingly, the assembly into inter-molecularly bonded macromolecular structures is known to reduce the recorded signal. For this type of assay, FTIR, either in solution or in the ATR mode, are more suited, especially in conditions where insoluble amyloids are expected to form.

Reviewer #3 (Remarks to the Author):

The manuscript by Horváth et al. is an interesting study on pH-dependent amyloid formation by peptide fragments from six important peptide hormones from the glucagon family. The authors use circular dichroism to monitor changes in the secondary structure of the peptide fragments over a range of pH's and find that most exhibit a shift from unstructured at neutral/high pH to β structure at low pH. The authors elucidate the crystal structures of the amyloid fibril-like assemblies for four of the six fragments and find that variation in crystallization components leads to polymorphism.

Overall, this was a well-designed study with high quality X-ray crystallography data and reasonable conclusions and interpretations of the X-ray data. The X-ray crystallography is the highlight of the manuscript and I commend the authors for their efforts in elucidating these structures. Furthermore, the manuscript is well-written and clearly communicates the findings and conclusions.

I do have some concerns about the CD data, though. While CD can provide a metric for monitoring the transition from an unstructured peptide to a folded beta sheet, I find these data alone unconvincing that the peptides form amyloid-like fibrils in solution at low pH and that the fibrils then reverse to their monomeric state at high pH.

Major concern with CD: The authors do not acknowledge the large spectroscopic contribution that Trp residues can have to CD spectra. Typically, Trp shows a positive band in the 220-230 nm range but can also contribute to other areas of the CD spectrum. Since each peptide studied contained, I question the reliability of using CD spectroscopy as the sole metric for fibril formation and monitoring the change from unstructured to beta structure under different pH's. While I agree that the structures of the peptides are changing when going from high to low pH, I am unconvinced based solely on CD data that the peptides are forming amyloid-like fibrils in solution.

The authors must better establish that the peptides are forming amyloid-like fibrils under the conditions in which CD was performed (i.e. low concentration/aqueous solution not the crystallographic conditions) at low pH using TEM or AFM. And then demonstrate that these amyloid-like fibrils "reverse" to their monomeric state or disappear at higher pH.

Minor concern with CD: I appreciate the experimental rigor of including many time points for each pH, but as presented in the extended figures, the CD data is difficult to meaningfully analyze and interpret. The panels that show spectra at different pH's in panel I for extended figures 2-6 begin to present the data in a more interpretable way, but why is the data from 168 hours shown and not 4 hours or 24 hours? This reviewer would like to see a CD graph for each peptide like the graph in panel I at the 4-hour time point (which should be plenty of time for the peptides to convert to beta structure) with only the pH ~2 and pH ~7 spectra shown.

The authors must also include characterization data (HPLC trace for purity, mass spectrum for identity) for each peptide presented in the paper.

Polymorphic Amyloid Nanostructures of Hormone Peptides Involved in Glucose Homeostasis: Designed for Reversible Amyloid Formation

point-by-point response to the reviewers

Reviewer 1-----

We thank Reviewer 1 for the thorough review of our manuscript and the helpful comments and suggestions.

R1.1.

One major limitation is the technique used to measure amyloid formation. The authors claim that “The aggregation propensities of hexapeptides derived from the gastrointestinal hormones were monitored by Circular Dichroism (CD) spectroscopy in the far-UV range.” No, CD monitors secondary structure changes and at best indirectly follows aggregation. And they do not provide any direct evidence for claims such as “the solubility of the hexapeptides is enhanced to such an extent that the equilibrium gets shifted toward the disassembled and solvated monomeric forms” – there are no solubility data. Aggregation can be measured by determining the solubility of the peptides (e.g. absorption before and after centrifugation to remove insoluble species) while amyloid formation can (to some approximation) be determined with ThT binding (though it is not quantitative in the case of e.g. polymorphism and pH changes can also affect it). The authors do not provide any direct evidence for amyloid formation under any conditions and this must be addressed carefully and thoroughly, not necessarily in all the data in Ext. Fig. 2-7 but certainly to a representative extent using methods above or equivalent alternatives.

Author1.1.

We thank the Reviewer for this comment which prompted a series of additional experiments and provided a substantial sounder basis for conclusions. We agree with the Reviewer that CD only monitors changes in the secondary structure. However, in the case of our hexapeptides, which lack tertiary structural elements in their monomeric forms that could shield any evolving nascent beta-edges, aggregation is highly likely if beta-strands carrying amyloidogenic sequences are formed. Therefore, we believe that the appearance of beta secondary structure in our case indicates amyloid aggregation as well. Although we did not provide exact solubility data, the ECD measurements incorporate these. The molar ellipticity values provided are adjusted based on the measured concentration of the APRs, which is determined using nanodrop at each individual sampling point (data not shown). However, to reaffirm our conviction and comply with the suggestions of the Reviewer, we performed ThT binding assays at both pH 4 and 7, included FTIR spectra of the proposed amyloid formation and we also obtained atomic force microscopy images of the fibrillar structures formed under acidic conditions. These analyses were carried out in an identical manner for all six studied APR hexapeptides and are summarized in Figure 2, and are shown and discussed in detail in the Extended Figures section (Figures 2, Extended Figures 2-7) – supporting our previously formed conclusions.

R1.2.

While I think the authors have a good case regarding pH-driven aggregation (subject to the extensions described above), the data are currently based on rather qualitative comparison of a large group of different CD spectra. It would improve the message considerably if the authors could encapsulate the observed changes in terms of extent of beta-sheet formation, e.g. by deconvoluting the spectra using the basis spectra in Fig. 2A and making a sensible estimate of amyloid (beta-sheet) formation as a function of pH from the many pH-varying spectra in Ext. Fig. 2-7. This could then be plotted in panels A of Ext. Fig. 2-7 plus Fig. 2CD.

A1.2.

Although the primary goal of ECD screening was to provide evidence of pH-dependent secondary structure conversion of the studied systems, we did consider providing deconvolution data of the measured spectra because changes in secondary structure would indeed be easier to interpret this way. However, the databases behind deconvolution methods are generally spectra of globular proteins instead of peptides. In the case of peptides- due to their short length - non-conventional spectral components (such as the signal arising from aromatic interactions) can have great (or even definitive) contributions, whereas in globular proteins, the contribution of well-known secondary structural elements dominates the ECD spectra. This creates instabilities in the predictions concerning peptidic systems, which were really amplified in the case of our very short APRs. As an example, we show here several deconvolution results analyzed quantitatively by BestSel. (<https://bestsel.elte.hu/> - Micsonai et al. PNAS 112:E3095-103 2015). We chose some characteristic ECD curves (a) and analyzed these spectra with the BeStSel program to determine their secondary structure content (α -helix, antiparallel β -sheet, parallel β -sheet, turn, and random structures) (b).

The results interpreted the spectra of those states that we considered as “amyloid” as containing a considerable beta-sheet content (parallel + antiparallel), with a predominance of antiparallel structure, consistent with the newly provided FTIR results and the crystal structures. However, the finer details of the deconvolutions remain hard to interpret.

- In the case of ECD curves with different spectral properties (EFAIWL, pH: 5.95, 1h (red line) - LYIQWL, pH: 7.6, 0.25h (grey dotted line)), we observed a nearly identical percentage of the antiparallel component according to the deconvolution data, however, EFAIWL does not exhibit the typical B-type properties at 205 nm.
- On the other hand, in a case when the deconvolution resulted in almost identical percentages for antiparallel content (53% and 56%), we still observed a significant difference between the ECD spectra (DFINWL, pH: 3.2, 1h (orange line) - DFVQWL pH: 3.18 1h (pink line)).
- For the LYIQWL peptide, based on the spectra recorded after 15 minutes of stirring, the program predicted 100% antiparallel β -sheet when the "transient" B-type ECD curve appeared (olive dotted line). However, further mixing (1h spectrum) resulted – according to the deconvolution results in a decrease to 34.4% antiparallel β -sheet content (green dotted line). In our view, the only difference between the two spectra is the disappearance of the assigned aromatic couplet near 235 nm, in the case of the latter.

Based on the above-mentioned examples, we concluded that the determination of secondary structure contents with deconvolution does not simplify the interpretation of the results. Although the

method is able to provide an approximate secondary structure composition consistent with the formation of amyloid nanostructures, we do not believe that the presentation of these results in the manuscript would illustrate our points more effectively.

R1.3.

All the data above are from hexapeptides rather than the peptide hormones themselves which are the important biological species. How reversible is their amyloid formation with pH changes? Can the authors address this experimentally?

A1.3.

Maji et al. (<https://doi.org/10.1126/science.1173155>.) previously demonstrated experimentally the reversibility of the amyloid formation of some hormone peptides (beta-endorphin, ACTH-beta-endorphin, arginine-vasopressin, somatostatin, prolactin) that are stored as amyloids. However, the structural mechanism behind the fibril reversibility was not proposed at that time. Seuring et al. (<https://doi.org/10.1038/s41594-020-00515-z>) were the first to propose a mechanism in the case of beta-endorphin, suggesting that the reversible nature of the fibril formation may be attributed to a protonable/deprotonable sidechain of the APR segment. Our study aimed to provide further examples of this pH-dependent switch mechanism in the case of the APRs of secretin-like hormone peptides and to highlight general concepts and evolutionary conserved features. While we believe that conducting additional experiments on the reversibility of the full-length peptides for all systems studied in this work is beyond the scope of this article, we carried out additional experiments using full-length glucagon to illustrate the applicability of our results. These new experiments demonstrate that the reversibility and pH dependence of the amyloid-formation process which was characterized using the APR segment is also present in the case of intact glucagon (see Extended Figure 8).

Minor issues:

R1.4.

Fig. 2B: it is unclear what the state "T" refers to. The authors refer to a "transition structure" but this is a vague term. The spectra referred to as T in Ext. Fig. 7D-H (orange brown) should first of all be clarified as being 15-min spectra and secondly it seems that they are combinations of the beta and beta' spectra shown in Fig. 2A – so could they not just be a mixture of these two states?

A1.4.

Thank you for pointing this out. The term "transition structure," is indeed vague, so we have corrected it to "transient signal," which might be a better description. We have also emphasized the legend of the transient signal both in the main text and the figure description. Although the transient signal may be a combination of the beta and beta' spectra, we still consider it transient. However, we believe that the provided reference (Wu, L.; McElheny, D.; Takekiyo, T.; Keiderling, T. A. Geometry and Efficacy of Cross-Strand Trp/Trp, Trp/Tyr, and Tyr/Tyr Aromatic Interaction in a Beta-Hairpin Peptide. *Biochemistry* 2010, 49 (22), 4705–4714. <https://doi.org/10.1021/bi100491s>.) more accurately describes the observed spectral features rather than the results of the deconvolution. In answer A1.2 we provided a deconvolution of this transient signal, which is predicted to be 100% antiparallel in character.

R1.5.

pH reversibility (ext. figures 2-6 panel J): It would help the reader if the authors also showed the pH 7.4 spectrum of each peptide when dissolved directly at pH 7.4 (i.e. without the amyloid "detour").

A1.5.

We have edited the implicated figures as the reviewer recommended for us. (Extended Figure 2-7. Panel b)

R1.6

I think the authors have a very good case for reversibility of peptide structure, but it would be good if they could show complete regain of solubility when reverting to pH 7.4, e.g. by comparing spectra before and after centrifugation.

A1.6.

We conducted a solubility experiment with the EFAWL hexapeptide by dissolving the lyophilized peptide at two different pH values and monitoring the solubility of the peptide over time, including during the process of amyloid fibril formation and the subsequent reversion.

Boxplot representation of solubility measurements by Nanodrop. The plain lined boxes indicate the measurements of non-centrifuged samples, while the dotted boxes show the concentrations measured after centrifugation at 13000 RPM for 1 minute. Each sample was measured five times. a) The concentrations of the initial and b) 72-hour agitated samples were measured at neutral pH. c) To test the effect of pH on solubility, the pH was adjusted from neutral to acidic after taking out half of the initial volume. (The added volume (15 μ l) of the 0.1M HCl was insignificant to the whole volume (1ml) of the sample) A significant decrease in concentration indicates instant precipitation. d) After 3 days of agitation, the measured concentration exhibited greater standard deviation values due to light scattering from the precipitated particles. However, the centrifuged sample showed a more consistent concentration range. e) By converting the pH of the aggregated sample back to neutral, the concentration of the initial sample was regained.

R1.7

“polymorph” => “polymorph” in several of the figures (e.g. Fig. 5); similar for “amiloid” in the text

A1.6.

We made the correction of these grammar mistakes.

Reviewer 2-----

We thank Reviewer 2 for the detailed analysis of our results and the constructive comments and suggestions.

R2.1

Although the authors devoted significant effort to solving eight different crystal structures of peptides belonging to the same or different hormones, and their models are interesting from a chemical point of view and can provide clues for designing reversible assemblies, I am not convinced that these short peptides provide new relevant information of biological interest in a context in which the amyloid structure of full-length amyloid hormones, as IAPP or glucagon itself (<https://doi.org/10.1101/2022.11.21.517306>) have been solved by Cryo-EM, in this later case in acidic conditions, where the assembly is permitted. Indeed, even if it is true that the DFVQWL segment of glucagon is involved in the central steric zipper that sustains the inner part of glucagon fibril, the contacts it establishes are different from the equivalents ones shown in the crystal states of the different peptides, essentially because, in the Cryo-EM structure, residues out of this region interact with this segment providing additional stability to the axis of the fibril. In addition, an important role is played by two positive residues adjacent to the segment studied here. Indeed, the presence of cationic residues adjacent to the hexapeptide seems to be a conserved feature of the family, and its inclusion in the studied peptides would have been relevant.

A2.1

Firstly, we would like to thank the referee for bringing to our attention the paper on parallel assembled glucagon fibrils solved by cryo-EM. However, we must clarify that this article had not been published at the time when we first submitted our paper to Nature in August 2022. Furthermore, the parallel-ordered example of glucagon amyloid cited by the referee, along with the antiparallel example of glucagon amyloid referenced by us (doi.org/10.1038/s41594-019-0238-6) support our hypothesis of possible polymorphism concerning the amyloid form of secretin-like hormone peptides.

We also wish to address the concern that „these short peptides may not provide new information of biological relevance in the context of the amyloid structure of full-length amyloid hormones”. We note that Eisenberg et. al. proposed the idea almost two decades ago, that APR regions of full-length proteins might drive the fibril formation of these proteins (<https://pubmed.ncbi.nlm.nih.gov/17468747/>). Several important properties of amyloid formation of full-length proteins could be explained by studies of the aggregation processes and amyloid structures of APR peptides despite the fact that neighboring residues indeed contribute to the formation of their amyloid forms. In our case, by studying APR sequences we were able to provide a

<https://doi.org/10.1101/2022.11.21.517306>

<https://doi.org/10.1038/s41594-019-0238-6>

feasible mechanism of the pH-dependent reversibility of peptide hormones and also shed light to the dual role of their receptor binding surfaces. Prior to our work, there was no experimental evidence for this. Furthermore, since apart from the two glucagon fibril structures, no structural data of the amyloid form of class B GPCR ligands has been determined to date, our study provides the first (admittedly limited) structural information about them.

In regards to the Reviewers' concern on „if it is true that the DFVQWL segment of glucagon is involved in the central steric zipper that sustains the inner part of glucagon fibril, the contacts it establishes are different from the equivalents ones shown in the crystal states of the different peptides, essentially because, in the Cryo-EM structure, residues out of this region interact with this segment providing additional stability to the axis of the fibril,, we believe that although the structure of amyloid fibrils of full-length glucagon inevitably differs from that of small APR sequences, important factors of the presented structures of homologous APR regions do resemble greatly the mentioned cryo-EM and ssNMR structures. Two such examples are provided below:

- 1) Antiparallel assembly (Extended Figure 12): Although we were not able to obtain measurable crystals containing the APR^{gluc} peptide, the antiparallel structure of the ordered state of APR^{tc5b} (LYIQWL), the sequential homolog of DFVQWL, exhibits a similar scheme of contact surfaces. Specifically, there are odd (C,D) and even (E,F) surfaces where identical contact is formed between the sidechains facing each other as is seen in the antiparallel amyloid of full-length glucagon.

- 2) Parallel assembly (Extended Figure 13): The parallel form of the DFINWL, which is an even closer sequential homolog of DFVQWL, has a similar structure of shifted contact surfaces between even-numbered residues F-L, Q-Q, and N-N, as was seen in case of the parallel amyloid structure of DFVQWL, referenced by the Reviewer. The only difference between the two is the interaction between Q-Q and N-N, which is also quite similar due to their comparable chemical nature.

We cannot completely confirm that the out-of-APR segments, particularly the N-terminus, interact with the APR segment on the opposite side of the buried steric zipper. However, upon examining the cross-section of the parallel Glucagon fibril, we indeed noticed only one possible interaction between Trp and the Met5 of the N-terminus, which may provide extra stability.

We completely agree with the Reviewer's view regarding the significance of positively charged residues that flank the APRs. We believe that these residues play a crucial role in fine-tuning both amyloid formation and its reversibility. However, it is important to note that while the sequence of the APR region is highly conserved, the Arg and Lys residues are not. For instance, in GLP-1 and exenatide, these residues are present on both sides of the APR region, whereas, in GLP-2, they are only present on the preceding side. On the other hand, in GIP and Glucagon, these residues are absent altogether.

Although in the present work, we focused on the role of the highly conserved receptor binding sequences in amyloid formation, as a continuation of this work we also plan to study the role of charged gatekeeper residues flanking APR regions.

R2.2

From the sentence, "This reversibility is triggered by the complete protonation of the C-terminal carboxylate of LYIQWL, yielding a monopole state with a positively charged N-terminal. In contrast to the wild type sequences discussed above." and the protonation graphs, it is deduced that in the used peptides, the N-terminal and C-terminal ends are not protected, and thus, the peptides bear two additional charges that respond to the pH but that are not present in the context of the natural sequence. This is surprising, since it is a common practice in the amyloid field to protect such ends to discard their potential contribution to the conformational conversion, especially when the net charge of the sequence is relevant for the experiments, as this is the case. This, together with the fact that the conditions used in crystallography are clearly far away from physiological ones, either extracellular or intra-secretory granules, including the use of 30% of organic solvent and pHs as low as 2-3 using TFA, argue against the observed structures and interactions occurring in a biological context.

A2.2

We agree with the Reviewer that protecting the N-terminal and C-terminal ends would bring us closer to naturally occurring conditions. However, with this approach, managing the synthesis of peptides and handling experiments would become difficult due to the low or no solubility of the peptides in aqueous media. Using the unprotected-backbone approach, we were able to conduct our experiments in water or water-organic solvent mixtures. Crystallographic studies of amyloid-forming peptides were performed almost exclusively on unprotected peptides in the past, as illustrated by the more than 100 such structures in the PDB. As we wanted to use the same peptides for both crystallographic studies (where higher concentrations may be required) and ECD measurements to obtain comparable results, we decided to use unprotected peptides for this study.

We note that crystallization of amyloid-forming peptides and other globular proteins in general usually requires non-physiological conditions (high concentrations of PEGs, salts, additives, organic solvents, various buffers...). Specifically, over 100 amyloid crystal structures of short peptides can be found in the PDB database that were crystallized under non-physiological conditions (often in the presence of organic solvents). These structures, however, were successfully used for understanding and classifying the basic topologies of amyloid structures, and also for explaining several specific aspects of the amyloid formation of physiologically relevant full-length proteins (eg. reversibility in the case of LARKSs (<https://www.ncbi.nlm.nih.gov/pmc/articles/PMC6192703/>), cytotoxic amyloid formation due to mutations in the sequence (<https://www.ncbi.nlm.nih.gov/pmc/articles/PMC5207774/>)), thereby clearly illustrating the biological relevance of such structures. It is the accepted paradigm that the basic interaction types, centers, and topologies (synthons) that govern the interaction of the shorter monomers define the amyloid buildup of longer segments too, attenuated by – naturally – the steric and electrostatic complementarity of the surrounding residues. And since APRs were also shown to play a crucial role in especially the initial steps of amyloid formation, we feel that our approach is justifiable.

Although we tried to crystallize all peptides in conditions close to the conditions of the ECD measurements, it was possible only in two cases (those of APR^{ex-4} (LFIEWL) and APR^{tc5b} (LYIQWL)). The addition of co-solvents or buffering agents proved to be necessary for growing high-quality single crystals of the other sequences.

It is also important to note that glucagon, as a medicine, is formulated to be dissolved in a solvent with a pH of 2-3 before injection, making low pH conditions relevant from the pharmaceutical perspective. Moreover, the solution must be made *in situ* just right before the administration, to avoid the possible aggregation at this pH. This example underscores the importance of considering the pharmaceutical perspective, not just its physiological relevance, as we have pointed out in our conclusion.

To study whether terminal protection of the peptides has a significant effect on the structure and mechanism of formation of amyloid nanostructures, as an example we synthesized the N-terminal acetyl-protected variant of EFlAWL (Ac-EFlAWL). The Ac-EFlAWL peptide was successfully crystallized and we determined its crystal structure. The determined new structure is very similar to that of pEFlAWL which was already presented in the original manuscript. We note that although this new structure contains Glu instead of pGlu, besides being N-terminally protected, the topology of the steric zippers it forms also belongs to class 7 (similarly to pEFlAWL) and similar interaction surfaces could be identified in the two (Figure 4). Furthermore, the ECD spectra also exhibited identical spectral features in both forms of EFlAWL (Supplementary Figure 5-6). Thus the Ac-EFlAWL system was a highly useful addition to our dataset – we are grateful to the Reviewer for prompting its synthesis and analysis. We believe that these results indicate that side chains of the studied hexapeptides define the main aspects of their amyloid assembly and protection of termini might not have such a significant structural effect.

R2.3

Unexpectedly, the amyloid nature of the structures at the different pHs used to monitor secondary structure using CD is not demonstrated. Techniques like Thioflavin-T and Congo Red binding/staining (or alternative amyloid-dyes), light scattering, and Transmission Electron Microscopy are conventional in the field and should have been used. In their absence, the formation of an inter-molecular β -sheet of amyloid nature at around neutral pH is arguable.

A2.3

To provide further evidence for the amyloid formation of the studied hexapeptides, we performed Thioflavin-T assay and FTIR measurements for all peptides at both acidic and neutral pHs. These results confirmed the amyloid nature of the aggregates we observed by ECD spectroscopy. We also performed AFM experiments, which showed fibrillar nanostructures characteristic of amyloid formation at acidic pH. (pH = 4.2) These are summarized in Figure 2. and shown in detail in Extended Figures 2-7 (Figure 2, Extended Figures 2-7). We believe that these new results together with our previous ECD and X-ray crystallographic data provide sufficient experimental evidence for the pH-dependent amyloid formation of the studied hexapeptides.

R2.4

The protonation state of the different pHs is calculated theoretically from the pKa of the individual residues in the sequence. However, it would have been more correct to calculate it experimentally in the different solutions at the peptide concentrations used in the CD experiments, for example, by measuring the sample's Z-potential.

A2.4

In our opinion using theoretically calculated pKa values instead of experimentally determined ones for hexapeptides is acceptable, because these peptides are solvated and unstructured in their monomeric forms, lacking any sidechain interactions that may influence the pKa values of the protonated group. As far as we know, Z-potential titration can determine the overall charge of the

peptides, but it cannot measure individual micro-constants of the protonable functional groups. Therefore, if two pKa values such as the C-terminal and the carboxylic sidechain of Asp or Glu are close enough, it may not be possible to differentiate between them using this method. To overcome this issue, NMR spectroscopy can be used since it can accurately determine individual pKa values by observing changes in chemical shifts as a function of pH. However, NMR spectroscopy requires the full assignment of all six hexapeptides, and due to their aggregation propensity, measuring these values accurately in an acidic pH range may not be feasible. It is also important to note that the aggregation propensity of the peptides may interfere with the accurate determination of their overall charge using Z-potential titration.

R2.5

The assignment of the different conformational states, unfolded and beta-sheet, by CD is not convincing since most spectra are not canonical, not displaying the typical signatures of this type of secondary polypeptide conformations. This is normal for such short peptides with high aromatic content strongly influencing the UV-spectrum. Moreover, CD is not the best technique for measuring peptides of low solubility, such as those in this study, because it is only sensible to the peptide in solution. Accordingly, the assembly into inter-molecularly bonded macromolecular structures is known to reduce the recorded signal. For this type of assay, FTIR, either in solution or in the ATR mode, are more suited, especially in conditions where insoluble amyloids are expected to form.

A2.5

We are grateful to the referee for recommending FTIR experiments in ATR mode, which we performed at a higher concentration than the ECD measurements due to the lower sensitivity of FTIR. We conducted a full cycle of amyloid formation and reversibility experiments on individual peptide samples, starting with neutral conditions and then shifting to acidic pH. After 24 hours of agitation, we reset the pH to neutral. At acidic pH, we observed an intense signal in the amide 1 region, specifically at 1625-1630 cm^{-1} , which is characteristic of the formation of extended intermolecular beta-sheets. As these results indeed complement greatly the somewhat ambiguous ECD spectra, we included the FTIR data in the manuscript (Figure 2, Extended Figures 2-7 – FTIR panels).

Reviewer 3-----

We thank Reviewer 3 for the helpful comments and suggestions.

R3.1

Major concern with CD: The authors do not acknowledge the large spectroscopic contribution that Trp residues can have to CD spectra. Typically, Trp shows a positive band in the 220-230 nm range but can also contribute to other areas of the CD spectrum. Since each peptide studied contained, I question the reliability of using CD spectroscopy as the sole metric for fibril formation and monitoring the change from unstructured to beta structure under different pH's. While I agree that the structures of the peptides are changing when going from high to low pH, I am unconvinced based solely on CD data that the peptides are forming amyloid-like fibrils in solution. The authors must better establish that the peptides are forming amyloid-like fibrils under the conditions in which CD was performed (i.e. low concentration/aqueous solution not the crystallographic conditions) at low pH using TEM or AFM. And then demonstrate that these amyloid-like fibrils "reverse" to their monomeric state or disappear at higher pH.

A3.1

We took into consideration the concerns raised by the Reviewers regarding the absence of complementary techniques. We carried out additional experiments, namely FTIR measurements, recorded AFM images, and monitored ThT binding-induced spectroscopic changes at different pHs, in the case of all considered systems. FTIR is a more robust technique for samples that are prone to aggregation and eliminates the possible interference of light diffraction that may affect ECD measurements. At acidic pH, we observed a strong signal in the amide 1 region at 1625-1630 cm^{-1} , which is indicative of the formation of extended intermolecular beta-sheets. Furthermore, upon resetting the pH to 7, the characteristic beta-sheet signal disappeared, and we obtained the initial FTIR signal. (Figure 2, Extended Figures 2-7 – FTIR panels). Along with the FTIR measurements, we also recorded AFM images of the same samples, which clearly demonstrated the presence of amyloid nanostructures with a characteristic linear, rod-like morphology. (Figure 2, Extended Figures 2-7 – AFM panels) As the FTIR results demonstrated the reversibility, we did not find it necessary to examine the reversibility by AFM as well, however, we provide examples for APR^{GIP} (Extended Figure 3), APR^{Ex-4} (Extended Figure 5) and APR^{tc5b} (Extended Figure 7, Supplementary Figure 9). Furthermore, pH-dependent amyloid formation was also demonstrated by ThT measurements (Figure 2, Extended Figures 2-7).

R3.2

Minor concern with CD: I appreciate the experimental rigor of including many time points for each pH, but as presented in the extended figures, the CD data is difficult to meaningfully analyze and interpret. The panels that show spectra at different pH's in the panel I for extended figures 2–6 begin to present the data in a more interpretable way, but why is the data from 168 hours shown and not 4 hours or 24 hours? This reviewer would like to see a CD graph for each peptide like the graph in the panel I at the 4-hour time point (which should be plenty of time for the peptides to convert to beta structure) with only the pH ~2 and pH ~7 spectra shown.

A3.2

As there is significant early progress in the aggregation process, it would be indeed more straightforward to compare the ECD spectra of a typical unstructured spectrum with that of an amyloid one. We have made the recommended modifications accordingly and stacked the ECD spectra measured after 4 hours as a function of pH. (see Panel a of Extended Figures 2-7)

R3.3

The authors must also include characterization data (HPLC trace for purity, mass spectrum for identity) for each peptide presented in the paper.

A3.3

We have included the analytical characterization of each examined peptide in Supplementary Figure 11.

REVIEWERS' COMMENTS

Reviewer #1 (Remarks to the Author):

The authors have addressed my concerns well. I only have two additional trivial points:

1. Why use the term ECD when CD is sufficient?
2. Answer to issue A1.6 is excellent and should be incorporated into the article (currently not the case).

Reviewer #2 (Remarks to the Author):

While I acknowledge the authors's efforts in addressing my comments, I still might express my concerns regarding the relevance of peptide assemblies obtained under conditions far from physiological settings, especially when we already have the structure of fibrils formed by the full-length hormone under milder conditions. It is unfortunate that this structure emerged while the authors were attempting to publish their present work, but this is a situation that we often encounter.

Undoubtedly, the pioneering work of D. Eisenberg on peptides and steric zippers provided highly valuable insights into the molecular basis of amyloid contacts when other approaches seemed technically unattainable. However, I believe that although this approach is still suitable for addressing specific questions on the generic nature of the amyloid assembly and provides excellent templates for computational studies, it has significant limitations for establishing a biological context, as many crucial interactions upon which the "real" assembly relies cannot be accounted for.

While I understand that crystallization conditions necessarily differ from those employed for biophysical characterization, I still harbor concerns regarding the comparability of data obtained in such disparate microenvironments. Although the argument that glucagon is formulated to be dissolved at low pH before injection is valid for the pH parameter, it is worth noting that these formulations lack the high concentrations of cosolvents used in the crystallization process.

I commend the authors for their efforts in demonstrating the amyloid-like nature of the assemblies, which I find convincing.

In my opinion, performing Z-potential titration measurements would have been important to complement the theoretical protonation states. While it is evident that this technique does not provide

information at the residue level, based on our expertise, significant deviations from theoretical values can arise.

Reviewer #3 (Remarks to the Author):

This reviewer thanks the authors for addressing the comments and concerns brought forth in the initial review. This reviewer is satisfied with the additional experiments performed, and believes the new data strengthens the paper and the authors conclusions. This reviewer recommends publication without additional changes.

Polymorphic Amyloid Nanostructures of Hormone Peptides Involved in Glucose Homeostasis Display Reversible Amyloid Formation

Second point-by-point response to the Reviewers

Reviewer 1-----

We would like to thank for Reviewer 1 again for their constructive criticism, meaningful comments and recommendations during the entire revision process.

R1.1.

Why use the term ECD when CD is sufficient?

Author1.1.

Thank You for this simplification. We have now replaced the term throughout the manuscript and the supplementary material to maintain consistency.

R1.2.

Answer to issue A1.6 is excellent and should be incorporated into the article (currently not the case).

Author1.2

We are glad that Reviewer 1 found our earlier response regarding the concentration issue appropriate. We have included this graph in Supplementary Information File as Figure 17.

Reviewer 2-----

We highly appreciate the thorough work done by Reviewer 2 and the opportunity it has provided for further discussion.

R2.1.

While I acknowledge the authors's efforts in addressing my comments, I still might express my concerns regarding the relevance of peptide assemblies obtained under conditions far from physiological settings, especially when we already have the structure of fibrils formed by the full-length hormone under milder conditions. It is unfortunate that this structure emerged while the authors were attempting to publish their present work, but this is a situation that we often encounter.

Author2.1

We fully agree with Reviewer 2 concerning the fact that fibrils of full-length hormones obtained from physiological sources or grown at near physiological conditions provide a more complete picture of mature amyloids. Therefore, in case of glucagon - especially since we were unable to determine the atomic resolution structure of its APR segment - the recently deposited solid phase NMR (6nzn) and cryoEM (7xm8) structures provide unprecedented and significant detail of its aggregated phase. However, the structure of the mature fibrils does not necessarily provide information about the nucleation process of amyloid formation or about the structure of the first ordered oligomers that are formed by the most aggregation prone core segments. We hope that our results provide a glimpse into this realm. Furthermore we have successfully obtained the structural information for the ordering of the APR segments of all the other incretin hormones probed in this work, which currently represents the only available data on their amyloid form. In the Discussion, we introduced and described in detail both mature fibrillar structures of glucagon (6nzn, 7xm8) - as requested by

the reviewer - to provide context for our results. Comparative analysis revealed striking similarities between the steric zippers of the available fibrils and our crystal structures, however as pointed out by the Reviewer, our crystal structures may “not provide a representation of truly physiological settings”. However, our experimental findings, as well as the two fundamentally different, published structures of mature glucagon fibrils, support our proposal that polymorphism is a general characteristic of the aggregated phase of these hormone peptides. This hypothesis is further supported by the preprint on parallel glucagon fibrils (7xm8), published after our submission, which supports our conclusions. We think, it might actually be encouraging for the research community that two different groups applying such different approaches have reached the same findings.

R2.2.

Undoubtedly, the pioneering work of D. Eisenberg on peptides and steric zippers provided highly valuable insights into the molecular basis of amyloid contacts when other approaches seemed technically unattainable. However, I believe that although this approach is still suitable for addressing specific questions on the generic nature of the amyloid assembly and provides excellent templates for computational studies, it has significant limitations for establishing a biological context, as many crucial interactions upon which the "real" assembly relies cannot be accounted for.

Author2.2

As the reviewer rightfully pointed out, the mentioned approach is indeed limited. However, it is important to note that the deposited two glucagon structures are - strictly speaking – abstractions of the true biological context, since both of them were grown *in vitro* and not extracted directly from acidic vesicles. Considering that the experimental circumstances were similar, yet they yielded completely different assemblies, we might safely assume the "real" biological assembly may exhibit a completely different morphology or a mixture of the two determined ones. Nevertheless, we believe that these structures are of great value given the extremely limited source of structural information regarding functional amyloids available currently. They allow the detailed study of the topologies and interactions that stabilize these aggregated forms. Furthermore, all the examples mentioned above help us understand the generic process of amyloid formation and its reversibility.

R2.3. While I understand that crystallization conditions necessarily differ from those employed for biophysical characterization, I still harbor concerns regarding the comparability of data obtained in such disparate microenvironments. Although the argument that glucagon is formulated to be dissolved at low pH before injection is valid for the pH parameter, it is worth noting that these formulations lack the high concentrations of cosolvents used in the crystallization process.

Author2.3

With this example we did not intend to compare the similarities or dissimilarities of the amyloid forming conditions. Instead, we would have liked to emphasize to the reviewer that conditions differing from biological ones may also be important from another point of view, that may be as significant as the physiological relevance - that of the pharmaceutical industry, which continuously strives to mimic these naturally occurring molecules in search of more biocompatible and effective drug molecules.

R2.4. I commend the authors for their efforts in demonstrating the amyloid-like nature of the assemblies, which I find convincing.

Author2.4

We are delighted. We are also grateful to You for Your earlier recommendations concerning the experimental techniques that could be used to complete and complement our initial results.

R2.5.

In my opinion, performing Z-potential titration measurements would have been important to complement the theoretical protonation states. While it is evident that this technique does not provide information at the residue level, based on our expertise, significant deviations from theoretical values can arise.

Author2.5

We also agree that it would be important to determine the exact pK_a values for each individual functional group and this knowledge would improve our models. However, our experimental approaches are limited due to the pH-dependent self-aggregation of our samples. In order to reach a reasonable compromise, we employed the PropKa software to determine the individual pK_a values based on the structure. To achieve this, we utilized our crystal structures and extracted the missing, homologous APR segments from the fibrillar structures of glucagon. We incorporated these structure-based, computational pK_a values into the final version of the manuscript and subsequently recalculated the species distribution. To address this pK_a -related question, we have added a new subsection to the Methods and Materials section, as well as a Supplementary Discussion where we express our perspective on this matter.

Reviewer 3-----

We are grateful for the time and effort of Reviewer 3 dedicated to the correction of our manuscript. Your insightful comments and meaningful suggestions have greatly enhanced the quality of our work.